# An obesogenic *FTO* allele causes accelerated development, growth and insulin resistance in human skeletal muscle cells

Lu Guang[1,2,3], Shilin Ma[1,2,3], Ziyue Yao[1,2,3], Dan Song[1,2,3,4], Yu Chen[1,2,3], Shuqing Liu[1,2,3], Peng Wang[1,2,3], Jiali Su[1,2,3], Yuefan Wang[1,2,3], Lanfang Luo[1,4,5] & Ng Shyh-Chang ®[1,2,3,4] ✉

Human GWAS have shown that obesogenic *FTO* polymorphisms correlate with lean mass, but the mechanisms have remained unclear. It is counterintuitive because lean mass is inversely correlated with obesity and metabolic diseases. Here, we use CRISPR to knock-in *FTO*^rs9939609-A into hESC-derived tissue models, to elucidate potentially hidden roles of *FTO* during development. We find that among human tissues, *FTO*^rs9939609-A most robustly affect human muscle progenitors' proliferation, differentiation, senescence, thereby accelerating muscle developmental and metabolic aging. An edited *FTO*^rs9939609-A allele overstimulates insulin/IGF signaling via increased muscle-specific enhancer H3K27ac, FTO expression and m6A demethylation of *H19* lncRNA and *IGF2* mRNA, with excessive insulin/IGF signaling leading to insulin resistance upon replicative aging or exposure to high fat diet. This FTO-m6A-*H19/IGF2* circuit may explain paradoxical GWAS findings linking *FTO*^rs9939609-A to both leanness and obesity. Our results provide a proof-of-principle that CRISPR-hESC-tissue platforms can be harnessed to resolve puzzles in human metabolism.

Up to 58% of the global adult human population is predicted to be overweight by 2030[1], due to a mixture of environmental and genetic factors. Obesity is associated with increased body mass index (BMI) and insulin resistance (IR), as well as several comorbidities, including type 2 diabetes (T2D) and a cluster of Metabolic Syndrome (MetS) diseases. Genome-wide association studies (GWASs) have identified single nucleotide polymorphisms (SNPs) which are strongly associated with BMI, IR, obesity, T2D and the Metabolic Syndrome (MetS)[2–8]. However, *FTO* is almost always the outlier locus with the most robust associations with these conditions. In fact, *FTO* SNPs have some of the highest -log(P) values in the history of GWASs, including the well-studied rs9939609 SNP[3,4,9]. The minor allele frequency of *FTO*^rs9939609-A is 12–20% in individuals of East Asian descent, and ~40% in individuals

of European descent, who have a much higher risk of insulin resistance than individuals with the *FTO*^rs9939609-T allele[3,4,9].

Although the mechanisms that link these intronic SNPs to obesity and IR were not immediately obvious, some studies in mice suggested that *Fto* expression promotes obesity[10,11]. However, *Fto* knockout mice do not always display a lean phenotype[12,13]. In contrast to the confusing growth phenotypes in *Fto* mouse models, *FTO* knockout is simply lethal in humans during development, indicating a fundamental difference between mouse *Fto* and human *FTO* circuits[14]. Mice overexpressing *Fto* also have reduced leptin[10], unlike humans with high-risk *FTO* alleles[15–17]. These conflicting findings between mice and humans, and between genetic mouse models, suggest a renewed focus on human cells is necessary to better understand the biology of *FTO*.

[1]Key Laboratory of Organ Regeneration and Reconstruction, State Key Laboratory of Stem Cell and Reproductive Biology, Institute of Zoology, Chinese Academy of Sciences, Beijing, China. [2]Institute of Stem Cells and Regeneration, Chinese Academy of Sciences, Beijing, China. [3]University of Chinese Academy of Sciences, Beijing, China. [4]Beijing Institute for Stem Cell and Regenerative Medicine, Beijing, China. [5]School of Bioengineering, Zhuhai Campus of Zunyi Medical University, Zhuhai, Guangdong, China. ✉e-mail: huangsq@ioz.ac.cn

In human samples, there is some evidence that the high-risk A allele at $FTO^{rs9939609}$ leads to obesity by increasing $FTO$ expression in human fibroblasts and blood cells[7,18]. However, other studies have shown that the heritability of $FTO$ expression in human tissues is low[19–21]. Expression quantitative trait locus (eQTL) analyzes have failed to document any impact of the $FTO$ SNPs on $FTO$ levels in human cell-lines and tissues[22–25]. Most disease-associated SNPs, although often located in gene regulatory regions, do not have detectable or significant effects on adult tissue gene expression[26]. Despite the importance of developmental biology to disease, studies of human progenitor cells are infrequent, partly due to the inaccessibility of developing tissues. Many developmental events are transient or rare, making an in vitro hESC-tissue model platform[27] especially useful for studying SNPs and eQTLs that may manifest effects only fleetingly[28–30]. Such considerations highlight the potential value of generating hESC-tissue models that differ in just one influential $FTO$ SNP to clearly elucidate the mechanistic role(s) of the $FTO$ genotype.

FTO catalyzes $N^6$-methyladenosine (m6A) demethylation in an α-ketoglutarate- and $Fe^{2+}$-dependent manner[8]. FTO was shown to play a m6A-dependent role in adipogenesis, altering the alternative splicing of RUNX1T1 to modulate preadipocyte differentiation[31,32]. In addition, animal and human cell models suggest that $FTO$ SNPs can also alter the expression of adjacent genes such as $IRX3$, $IRX5$ and $RPGRIP1L$, to modulate adipocyte browning and hypothalamic neurons during IR[22,23]. Moreover, many $FTO$ SNPs are in linkage disequilibrium with dozens of other SNPs[33], confounding the causal role(s) of strong SNPs like $FTO^{rs9939609-A}$ in regulating the expression of $FTO$ and other neighboring genes during human IR.

Given the overall body growth phenotypes of $FTO$ SNPs, which involves other non-adipose tissues, it was also unclear if $FTO^{rs9939609-A}$ only affects adipogenesis to regulate human BMI or if it also affects other tissues that may contribute to the BMI. Skeletal muscle mass accounts for ~40% of the body weight, and contributes heavily to the BMI. Furthermore, muscle is a major site for insulin-sensitive glucose uptake. Thus, it is plausible that $FTO^{rs9939609-A}$ also regulates human BMI and IR via effects on human myogenesis, especially since $Fto$ is necessary for muscle regeneration[34–36], and human genetics studies have shown that $FTO^{rs9939609-A}$ is associated with greater lean mass[37]. However, the mechanisms underlying the positive association of human $FTO^{rs9939609-A}$ with both fat mass and lean mass have remained unclear. This association is counterintuitive and paradoxical, because studies have indicated that lean mass is negatively correlated with obesity, IR and MetS[38]. To solve this question of the tissue-specificity of human $FTO^{rs9939609-A}$, and questions regarding its mechanistic role in IR, we used CRISPR prime editing to knock-in $FTO^{rs9939609-A}$ into several isogenic $FTO^{rs9939609-TT}$ hESC lines to dissect the role of a single $FTO$ SNP during human tissue development. We found that among 5 tissue lineages tested, $FTO^{rs9939609-A}$ had the most robust effect on human skeletal muscle differentiation, growth, as well as IR and senescence, thereby accelerating the developmental age of muscles via increased FTO-$H19$/$IGF2$ signaling. These results may explain the paradoxical effects of $FTO$ on both lean mass and fat mass during obesity and IR in humans.

## Results

### CRISPR prime editing of one FTO SNP locus accelerates myogenic development

To model the influence of $FTO^{rs9939609-A}$ on human developmental fates, we used CRISPR/Cas9-based prime editing of $FTO^{rs9939609-TT}$ hESCs, followed by induction of their differentiation into mesodermal progenitors, endodermal progenitors and neuroectodermal progenitors (Fig. 1a). Firstly, we designed pegRNAs and transfected them along with Cas9 to generate targeted knock-in clones that harbored the obesity-associated A allele at rs9939609, as confirmed by PCR and genomic sequencing (Supplementary Fig. 1). Secondly, we expanded

these clones and verified their genomic stability (Supplementary Fig. 2a) and normal pluripotency (Supplementary Fig. 2b). In this way, we generated 6 different hESC clones with the $FTO^{rs9939609-A}$ SNP, i.e. both heterozygous and homozygous clones were derived from H1, H7 and H9 hESCs.

By subjecting these 6 clonal lines to stage-wise directed differentiation, we induced them to become paraxial mesoderm cells and dermamyotome-derived myogenic progenitors[39] (see Methods). Interestingly, relative to $FTO^{rs9939609-TT}$ hESC clones cultured in parallel, all 6 $FTO^{rs9939609-A}$ clones showed 50- to 350-fold higher expression of myogenic transcription factors, such as $PAX3$, $PAX7$, $MYF5$, $MYOD1$, and $MYOG$ ($P < 0.01$; Fig. 1b). Differentiation markers, such as myosin heavy chain $MYHC$ and skeletal muscle actin A ($ACTA1$), were also 100- to 150-fold higher than in $FTO^{rs9939609-TT}$ hESCs, suggesting that the $FTO^{rs9939609-A}$ SNP consistently promotes specification into myogenic progenitors with surprisingly high efficiency (Fig. 1b). In contrast, the $FTO^{rs9939609-A}$ SNP had less robust effects on adipose, cardiac, neural, and liver progenitor specification (Supplementary Figs. 3a–d), with only significant effects on two adipose specification markers $FABP4$ and $PPARG$ (Supplementary Fig. 3a), and a weaker effect on one neural specification marker $FOXG1$ (Supplementary Fig. 3c). While one $FTO^{rs9939609-A}$ cell-line (FTO-H1-2) did show increased adipogenesis (Supplementary Fig. 3e), the other $FTO^{rs9939609-A}$ cell-lines did not show such a robust effect.

By contrast, the $FTO^{rs9939609-A}$ allele exerted a strong and robust effect on skeletal muscle differentiation. By the time the myogenic progenitors were induced to commit to myocytes, the $FTO^{rs9939609-TT}$ hESCs-derived myocytes had narrowed their gap with $FTO^{rs9939609-A}$ myocytes with regard to myogenic factor expression, but the 6 clones with the $FTO^{rs9939609-A}$ SNP still showed an ~30-fold higher level of the muscle satellite cell (MuSC) transcription factor $PAX7$, an ~70-fold higher level of the myogenic transcription factor $MYF5$ and an ~50-fold higher level of the muscle commitment marker $ACTA1$ (Fig. 1b). Upon terminal differentiation into myotubes, $FTO^{rs9939609-A}$ myotubes not only showed higher levels of myogenic gene expression (Fig. 1b), but also higher levels of both type 1 slow-twitch myofiber markers and type 2 fast-twitch myofiber markers (Supplementary Fig. 4a). These observations were confirmed by RNA-sequencing, which showed that $FTO^{rs9939609-A}$ myotubes were significantly enriched in signatures for PAX3/MYOD-induced myogenic differentiation (Fig. 1c). By Western blotting we confirmed that all $FTO^{rs9939609-A}$ myotubes had higher levels of FTO protein, as well as PAX3, PAX7, MYOG (myogenin) and MHC (myosin heavy chain) protein expression, confirming that the $FTO^{rs9939609-A}$ SNP increased myogenic potential as a dominant allele (Fig. 1d). A different directed myogenesis protocol that was slightly less efficient produced similar results (Supplementary Figs. 4b–d).

### An edited FTO SNP accelerates maturation of myotubes and myofibers

Myotube terminal differentiation efficiency was further determined by the fusion index, based on anti-MHC staining at day 7. Compared with $FTO^{rs9939609-TT}$ hESCs-derived myocytes, $FTO^{rs9939609-A}$ myocytes differentiated into multinucleate myotubes to a higher degree (Fig. 2a). Morphological parameters such as myotube length and myotube area were also significantly enhanced in the $FTO^{rs9939609-A}$ myotubes (Fig. 2b). In addition, $FTO^{rs9939609-A}$ myotubes showed higher levels of contractility in 2D culture, a functional indicator of mature sarcomeres and thus myofiber maturation (Fig. 2c, Supplementary Movies 1-2). In contrast, when allowed to differentiate and mature into other human tissue models over longer periods of time, the $FTO^{rs9939609-A}$ allele still only had small and variable effects on the morphology of adipose, cardiac, neural and liver tissue models (Supplementary Figs. 3f–i). Both $FTO^{rs9939609-A}$ myotubes and $FTO^{rs9939609-TT}$ hESCs-derived myotubes could form 3D muscle organoids spontaneously (Fig. 2d, Supplementary Movies 3–4), but $FTO^{rs9939609-A}$ myofibers manifested higher contractility in response to depolarization by electrical stimulation

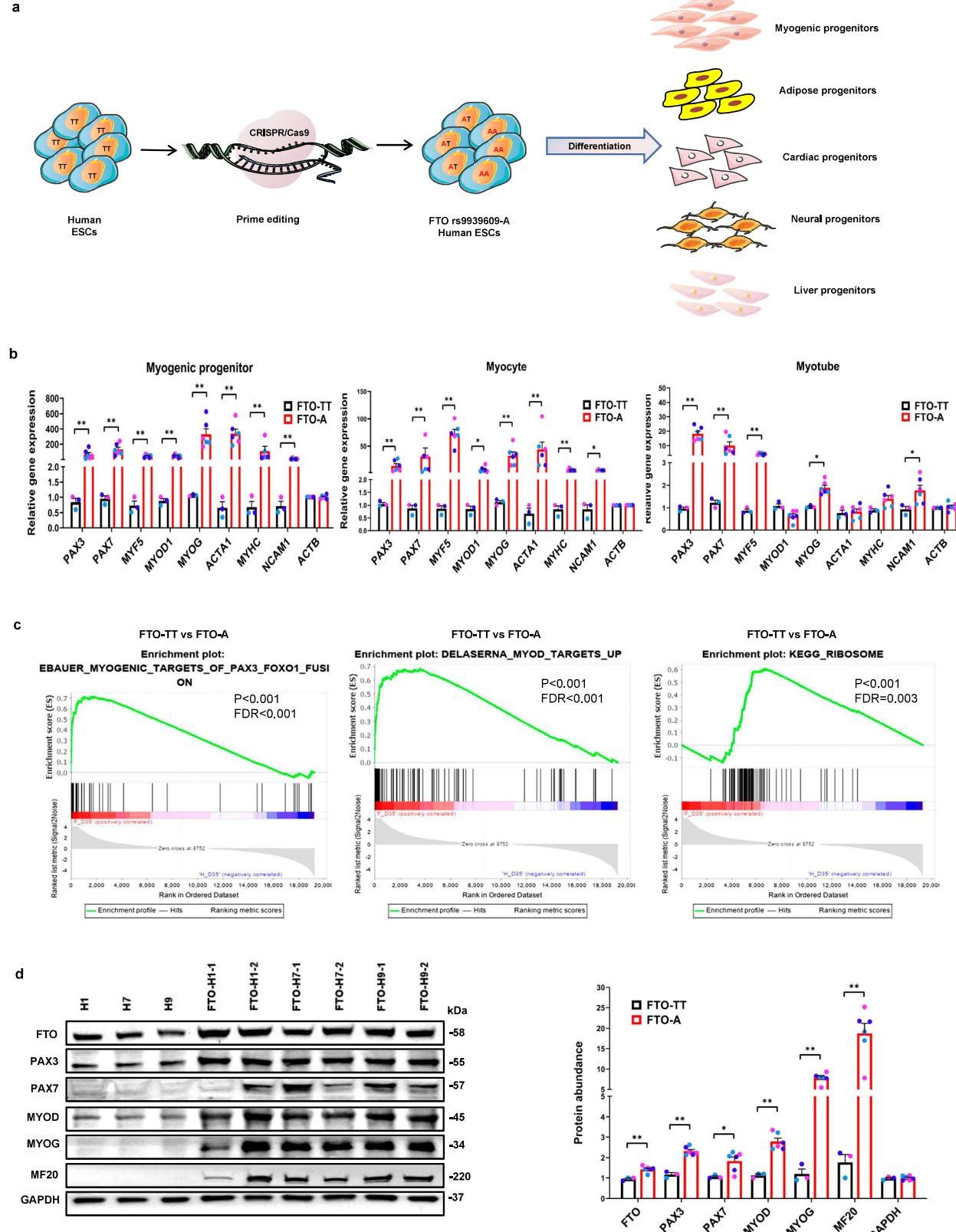

(Fig. 2e), and higher endurance for muscle contractions over long periods of time (Fig. 2f). Transcriptomic analysis with RNAseq showed that the 3D muscle organoids were significantly different in 1864 genes' expression ($P < 0.05$, Fig. 2g). Gene Ontology analysis of these differentially expressed genes revealed that $FTO^{rs9939609\text{-}A}$ muscle organoids showed significant upregulation of ion channels, neuromuscular synapse components, skeletal muscle differentiation genes, compared to $FTO^{rs9939609\text{-}TT}$ hESCs-derived muscle organoids (Fig. 2h). Detailed examination of slow-twitch type 1 and fast-twitch type 2 myofiber markers further revealed that $FTO^{rs9939609\text{-}A}$ accelerated the formation of both subtypes of myofibers (Supplementary Fig. 4a). In contrast, non-muscular genes such as genes implicated in early embryonic

**Fig. 1 | Amongst different lineages, $FTO^{rs9939609-A}$ promotes myogenesis most robustly. a** A schematic diagram illustrating our workflow utilizing CRISPR/Cas9-based prime editing of hESCs to study the effects of a specific *FTO* SNP on cellular differentiation. **b** Quantitative RT-PCR of myogenic markers in $FTO^{rs9939609-TT}$ and $FTO^{rs9939609-A}$ hESCs-derived myogenic progenitors, myocytes and myotubes. Paired box 3 (*PAX3*), Paired box 7 (*PAX7*), Myogenic factor 5 (*MYF5*), Myogenic differentiation 1 (*MYOD1*), Myogenin (*MYOG*), Skeletal muscle actin alpha 1 (*ACTA1*), Myosin heavy chain (*MYHC*), Neural cell adhesion molecule 1 (*NCAM1*), Actin Beta (*ACTB*), *n* = 3 biologically independent samples. **c** Gene Set Enrichment Analysis (GSEA) of the signatures enriched in $FTO^{rs9939609-A}$-myotubes, relative to $FTO^{rs9939609-TT}$

-myotubes, according to RNA-sequencing. **d** Left: western blot quantification of FTO, PAX3, PAX7, MYOD1, MYOG, MHC (myosin heavy chain, MF20) and GAPDH protein abundance in $FTO^{rs9939609-TT}$-myotubes: H1, H7, H9 and $FTO^{rs9939609-A}$-myotubes: FTO-H1-1, FTO-H1-2, FTO-H7-1, FTO-H7-2, FTO-H9-1, FTO-H9-2. Right: quantification of the abundance of myogenic differentiation proteins, *n* = 3 biologically independent samples. H1 hESC line colored in purple, H7 hESC line colored in light blue and H9 hESC line colored in dark blue. Data are presented as mean + SEM. Data were analyzed using Wald test (**c**). *P* values were calculated by two-tailed unpaired t-test. \**P* < 0.05, \*\**P* < 0.01. Source data are provided as a Source Data file.

morphogenesis, skeletal development, angiogenesis, cardiogenesis and extracellular matrix were downregulated (Fig. 2i). Thus, editing of an $FTO^{rs9939609-A}$ allele accelerated myogenic development and myofiber maturation.

## An edited *FTO* SNP promotes proliferation and senescence of myogenic progenitors

Given our observations that $FTO^{rs9939609-A}$ myogenic progenitors had higher levels of the muscle stem cell transcription factor *PAX7*, it was possible that $FTO^{rs9939609-A}$ myogenic progenitors had higher rates of proliferation. To confirm this possibility, we performed cell counting and found that $FTO^{rs9939609-A}$ expression did indeed promote a greater proliferation rate of human myogenic progenitors compared to the $FTO^{rs9939609-TT}$ hESCs-derived cells (Fig. 3a, b). An increase in proliferation could imply either improved self-renewal (and less senescence) or accelerated exhaustion of progenitor cells (and more senescence) during long-term culture and replicative aging. To test this, we performed senescence-associated β-galactosidase (SA-βgal) staining of the $FTO^{rs9939609-A}$ myogenic progenitors and found that they had about twice as many senescent cells as $FTO^{rs9939609-TT}$ hESCs-derived cells by passage 16 (Fig. 3c, d). These results suggest that the $FTO^{rs9939609-A}$ SNP accelerates the senescence and exhaustion of muscle progenitor cells after increasing proliferation.

## An edited *FTO* SNP accelerates high-fat diet serum-induced muscle insulin resistance

A profoundly significant association between the $FTO^{rs9939609-A}$ SNP and insulin resistance (IR), and thus the metabolic syndrome (MetS), is well-known to exist, but its causative mechanisms have remained unclear. Thus, we utilized our $FTO^{rs9939609-A}$ myotubes to model the interactions between genetic and environmental causes that lead to IR in humans with the $FTO^{rs9939609-A}$ SNP. To provide a readout of IR, we introduced the FoxO1-GFP fusion protein reporter into our cells as, under conditions of insulin sensitivity, PI3K-Akt phosphorylation of FoxO1 leads to a cytoplasmic localization of the FoxO1-GFP fusion protein reporter. Using this reporter system, we found that under conditions of serum starvation, or loss of insulin signaling, the FoxO1-GFP fusion protein is dephosphorylated and rapidly undergoes nuclear translocation (Fig. 4a, Supplementary Fig. 5a).

Using this model system, we performed high-throughput screening of human myotubes for different titrations and combinations of substances that promoted IR in the $FTO^{rs9939609-A}$ myotubes, relative to $FTO^{rs9939609-TT}$ myotubes. Exposure to different titrations of well-known IR inducers, such as palmitate, ceramide, IL-1b, IL-6 and TNFa, and their combinations, all failed to increase the IR of $FTO^{rs9939609-A}$ myotubes (Supplementary Fig. 5b), whereas exposure to serum from mice fed with a high-fat diet (HFD) robustly promoted IR after 14 days (Fig. 4b). This result held true for Zucker Diabetic Fatty Rat serum, as well (Supplementary Fig. 5c). By Western blotting, we also confirmed that the $FTO^{rs9939609-A}$ myotubes had significantly lower levels of phospho-IRS1, phospho-AKT and phospho-FOXO, which are other markers of insulin signaling, compared to $FTO^{rs9939609-TT}$ myotubes, after exposure to HFD serum (Fig. 4c). By glucose uptake assays, we confirmed these changes in insulin signaling led to

reduced insulin-stimulated glucose uptake (Fig. 4d). Thus, our results confirmed that an edited $FTO^{rs9939609-A}$ SNP can directly promote IR in human muscle cells, and that long-term exposure to a complex mixture of obesogenic molecules in HFD serum robustly elicits a muscle IR phenotype.

Interestingly, prior to exposure to HFD serum, $FTO^{rs9939609-A}$ myotubes had significantly higher levels of phospho-IRS1, phospho-AKT and phospho-FOXO, compared to $FTO^{rs9939609-TT}$ myotubes (Fig. 4e). This result suggests that $FTO^{rs9939609-A}$ myotubes actually have a higher degree of insulin sensitivity at baseline compared $FTO^{rs9939609-TT}$ myotubes, but the former cells are more sensitive to IR-inducing factors in the HFD serum. Along these lines, high-content imaging of $FTO^{rs9939609-A}$ cells after exposure to HFD serum revealed that these cells initially showed more FoxO1-GFP cytoplasmic localization at day 3 (Supplementary Fig. 5d), before showing a gradual increase in nuclear translocation frequency by day 14 (Fig. 4b). These results were confirmed with a functional assay for IR, involving serum starvation followed by insulin stimulation. At day 7 after exposure to HFD serum, phospho-AKT and phospho-FOXO levels were still similar between cells at the fed state (Fig. 4f). However, serum starvation showed that $FTO^{rs9939609-A}$ myotubes had higher levels of phospho-AKT and phospho-FOXO compared to $FTO^{rs9939609-TT}$ myotubes (Fig. 4f). Only after insulin stimulation, did we observe that $FTO^{rs9939609-A}$ myotubes were becoming less sensitive in AKT and FOXO phosphorylation, compared to $FTO^{rs9939609-TT}$ myotubes (Fig. 4f). Coupled with the results of both greater cell proliferation and accelerated exhaustion in $FTO^{rs9939609-A}$ myogenic progenitors, these paradoxical results suggest that $FTO^{rs9939609-A}$ cells express one or more factors which over-stimulate insulin/IGF signaling for myoblast proliferation and myotube insulin sensitivity initially, and then senescence and IR over time[40,41]. Interestingly, senescent hESC-myoblasts that can no longer proliferate also show IR, as measured by changes in FoxO1-GFP localization (Supplementary Fig. 5e).

## An edited *FTO* SNP causes an increase in *IGF2* and *H19* expression via m⁶A demethylation

A primary candidate for the factor that over-stimulates insulin/IGF signaling would be an IGF family member itself. Indeed, GSEA of the 3958 differentially expressed genes (P < 0.05) revealed by RNAseq of $FTO^{rs9939609-A}$ myotubes showed that an IGF signature was upregulated in $FTO^{rs9939609-A}$ myotubes, compared to $FTO^{rs9939609-TT}$ myotubes (Fig. 5a). Two plausible candidates for the insulin-like growth factors that may be expressed by the $FTO^{rs9939609-A}$ cells are IGF1 and IGF2. ELISA analysis for IGF1 levels showed no differences in IGF1 secretion between $FTO^{rs9939609-TT}$ and $FTO^{rs9939609-A}$ myotubes (Supplementary Fig. 6a), but showed that $FTO^{rs9939609-A}$ myotubes secreted 2-4X higher levels of IGF2 (Fig. 5b). By Western blotting, we further confirmed that $FTO^{rs9939609-A}$ myotubes expressed 4-5X higher levels of IGF2 protein, even without exposure to HFD serum (Fig. 5c). By gene expression profiling, we further found that *IGF2* mRNA levels more than doubled as well, in $FTO^{rs9939609-A}$ myotubes (Fig. 5d).

Moreover, FTO expression was tripled in $FTO^{rs9939609-A}$ myotubes (Fig. 5d). Scanning of databases revealed that $FTO^{rs9939609}$ was a potential eQTL, especially in the skeletal muscles (Supplementary

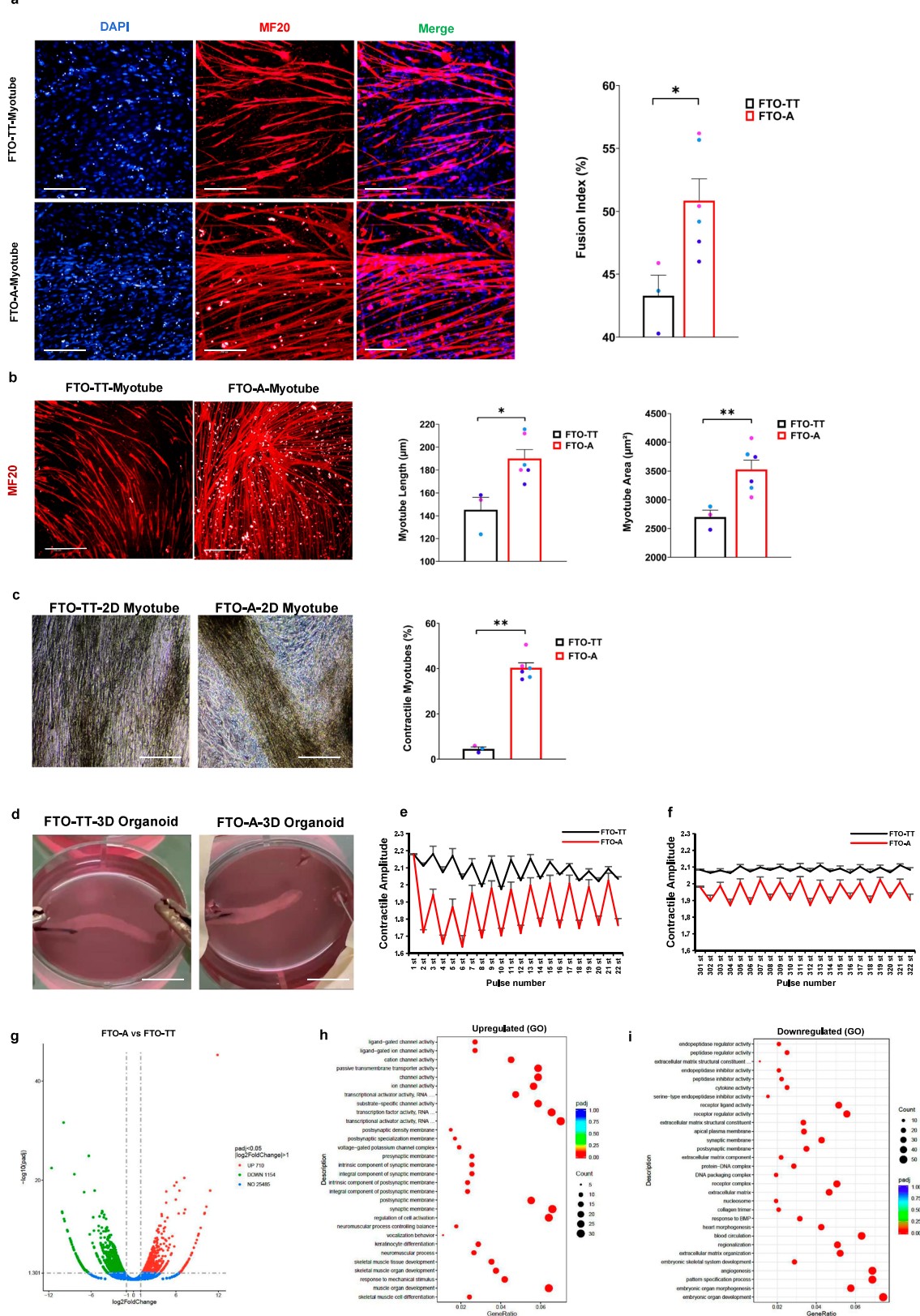

Figs. 6b, c). Close examination of the H3K27ac peaks near rs9939609 in the Encode database, revealed an FTO enhancer EH38E1816455 that lies within 1 kb of rs9939609, and it is a cCRE (cis-Regulatory Element) with peaks in DNAse hypersensitivity, H3K4me3 enrichment, H3K27ac enrichment, and CTCF binding, especially in muscle cells (Fig. 5e). In fact, a recent paper[42] showed a physical interaction between the FTO promoter and a nearby enhancer region encompassing rs9939609, and that the rs9939609-A allele correlated with higher *FTO* expression ($P = 0.011$), although no further molecular and cellular studies were performed. To experimentally determine if this FTO enhancer is affected by allelic differences in *FTO*[rs9939609], we performed H3K27ac ChIP-qPCR on all rs9939609-TT vs rs9939609-A lines, for

**Fig. 2 | *FTO^rs9939609-A* promotes myogenesis most robustly. a** Left: representative images of the *FTO^rs9939609-TT*-myotubes and *FTO^rs9939609-A*-myotubes. Myotubes as indicated by immunostaining for MHC (MF20; red). Nuclei were counterstained with DAPI. Scale bars, 200 μm. Right: percentage of fusion index of *FTO^rs9939609-TT*-myotubes and *FTO^rs9939609-A*-myotubes (FTO-TT vs. FTO-A, *P* = 0.02795), *n* = 3 biologically independent samples. **b** Left: representative images of *FTO^rs9939609-TT*-myotubes and *FTO^rs9939609-A*-myotubes. Myotubes as indicated by immunostaining for MHC (MF20; red). Scale bars, 200 μm. Right: morphological characteristics (myotube length and area) of *FTO^rs9939609-TT*-myotubes and FTO^rs9939609-A*-myotubes (FTO-TT vs. FTO-A, *P* = 0.01326, *P* = 0.00943), *n* = 3 biologically independent samples. **c** Left: representative images of 2D *FTO^rs9939609-TT* and *FTO^rs9939609-A* contractile myotubes. Scale bars, 200 μm. Right: percentage of contractility among *FTO^rs9939609-TT*-myotubes and *FTO^rs9939609-A*-myotubes (FTO-TT vs. FTO-A, *P* = 0.00001), *n* = 3 biologically independent samples. **d** Representative images of 3D *FTO^rs9939609-TT*-organoid

and *FTO^rs9939609-A*-organoid, *n* = 3 biologically independent samples. Scale bars, 8 mm. **e** The contraction process of the electrical pulse-stimulated 3D muscle organoids in (**d**) was recorded using a camera, the average contractile displacement was calculated by ImageJ software and plotted against the pulse number (1-22). **f** Average contractile displacement plotted against pulse number (301-322) in (**d**). **g** Volcano plot of differentially expressed genes in *FTO^rs9939609-A*-muscle organoids, relative to *FTO^rs9939609-TT*-muscle organoids. **h** Gene Ontology (GO) analysis of signatures upregulated in *FTO^rs9939609-A*-muscle organoids, relative to *FTO^rs9939609-TT*-muscle organoids. **i** Gene Ontology (GO) analysis of signatures downregulated in *FTO^rs9939609-A*-muscle organoids, relative to *FTO^rs9939609-TT*-muscle organoids. H1 hESC line colored in purple, H7 hESC line colored in light blue and H9 hESC line colored in dark blue. Data are presented as mean + SEM. Data were analyzed using Wald test (**g**) and hypergeometric test (**h**, **i**). *P* values were calculated by two-tailed unpaired t-test (a-c,e-f). *$P < 0.05$, **$P < 0.01$. Source data are provided as a Source Data file.

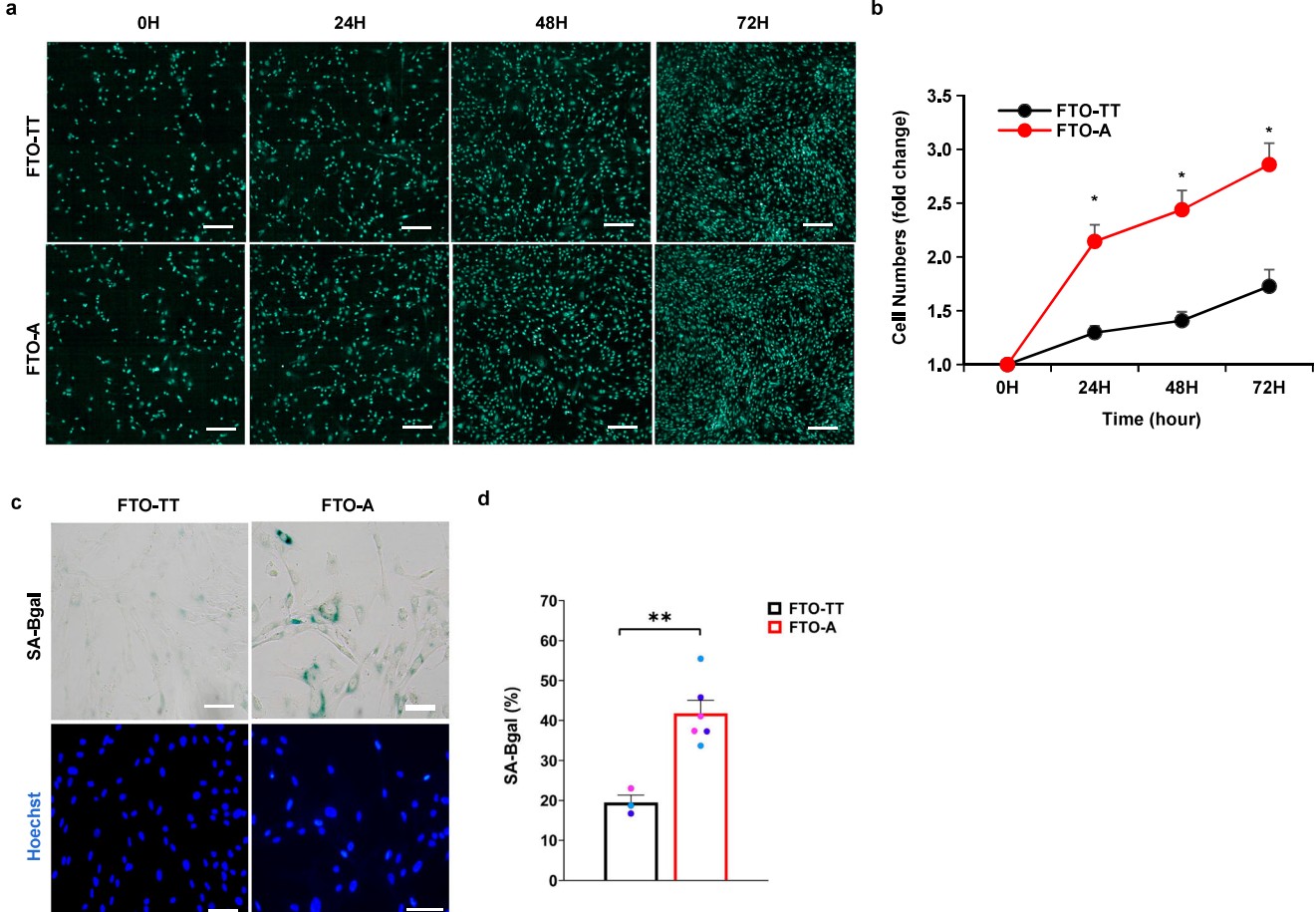

**Fig. 3 | *FTO^rs9939609-A* causes more rapid proliferation and senescence.**
**a** Representative images of *FTO^rs9939609-TT*-myoblasts and *FTO^rs9939609-A*-myoblasts that were automatically counted for Hoechst 33342+ cell numbers daily for 72 h using the Operetta high-content screening microscope. Scale bars, 100 μm.
**b** Quantification of fold change in myoblasts cell numbers, *n* = 3 biologically independent samples. **c** Representative images of senescent *FTO^rs9939609-TT*-myoblasts and *FTO^rs9939609-A*-myoblasts by passage 16. Senescent cells were stained for

β-galactosidase (SA-βgal). Scale bars, 100 μm. **d** Quantification of percentage of senescent *FTO^rs9939609-TT*-myoblasts and *FTO^rs9939609-A*-myoblasts (FTO-TT vs. FTO-A, *P* = 0.00250), *n* = 3 biologically independent samples. H1 hESC line colored in purple, H7 hESC line colored in light blue and H9 hESC line colored in dark blue. Data are presented as mean + SEM. *P* values were calculated by two-tailed unpaired t-test. *$P < 0.05$, **$P < 0.01$. Source data are provided as a Source Data file.

all 5 tissue types. Our results showed that the *FTO^rs9939609-A* mutation caused a significant increase in H3K27ac at this FTO enhancer EH38E1816455 in skeletal muscle cells (Fig. 5f). In contrast, adipocytes only showed a mild increase in H3K27ac at this FTO enhancer EH38E1816455, and little to no differences in other cell types (Supplementary Figs. 6d–g).

Motivated by the strengthening of FTO enhancer EH38E1816455 H3K27 acetylation and the tripling of mRNA expression of the FTO m6A demethylase in *FTO^rs9939609-A* myotubes, we measured and found

that the m6A methylation of *IGF2* mRNA was significantly lower by >2-fold in *FTO^rs9939609-A* myotubes (Fig. 5g). In contrast, the muscle expression of *IRX3 and IRX5* remained mostly unchanged (Supplementary Fig. 7), and *RPGRIP1L* could not be detected in muscle cells. These results suggest a unique mechanism for the *FTO^rs9939609-A* SNP, which increases a human muscle-specific FTO enhancer's activity, and thus human muscle FTO demethylase levels to facilitate the m6A demethylation of *IGF2* mRNA, which has been shown to increase mRNA stability and protein expression[43].

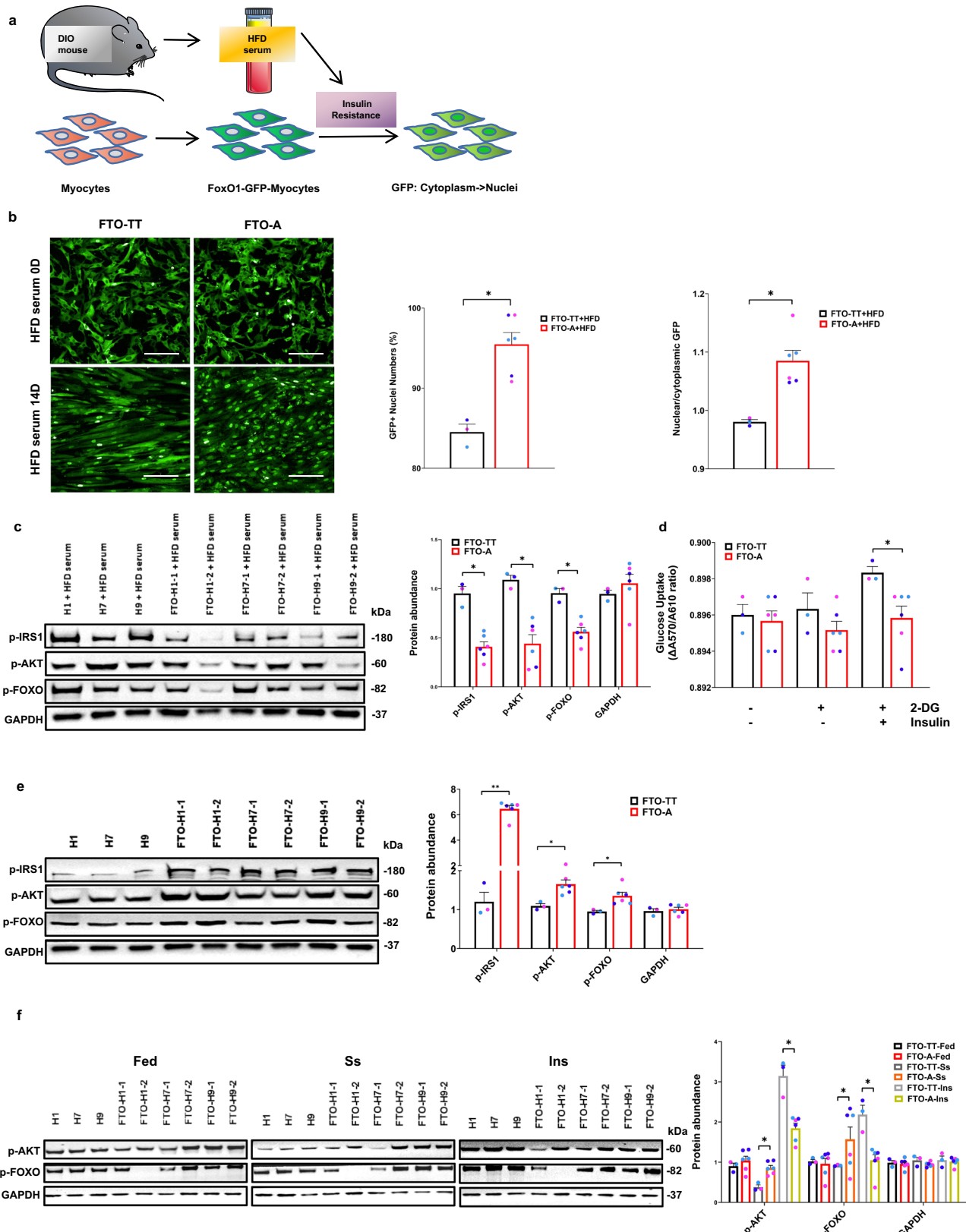

Unexpectedly, the *FTO^{rs9939609-A}* allele also dramatically increased the expression of the *H19* long non-coding RNA (lncRNA), a maternally inherited factor that is known to promote myogenesis and insulin/IGF signaling[44], by a few thousand-fold in myotubes (Fig. 5a, d). In addition, the m⁶A methylation of *H19* lncRNA was also significantly lower by >4-fold in *FTO^{rs9939609-A}* myotubes, compared to *FTO^{rs9939609-TT}* myotubes,

suggesting that the *FTO^{rs9939609-A}* allele increased *H19* lncRNA levels via m⁶A demethylation (Fig. 5g). To prove that the *FTO^{rs9939609-A}* allele increased *H19* and *IGF2* RNA levels via FTO-mediated m⁶A demethylation, we also performed siRNA knockdown and pharmacological inhibition of the FTO m⁶A demethylase. Our siRNA results showed that FTO is needed for the *FTO^{rs9939609-A}* allele to increase *H19* and *IGF2* RNA

**Fig. 4 | *FTO<sup>rs9939609-A</sup>* causes an increased propensity for HFD serum-induced insulin resistance. a** A schematic diagram illustrating the workflow by which HFD serum was tested for the induction of insulin resistance in the human myocytes model. **b** Left: representative images of FoxO1-GFP-infected *FTO<sup>rs9939609-TT</sup>*-myocytes and *FTO<sup>rs9939609-A</sup>*-myocytes exposed to 1% HFD serum at day 0 and day 14. Scale bars, 200 µm. Middle: quantification of percentage of FoxO1-GFP+ nuclei numbers at the end of day 14, $n = 3$ biologically independent samples. Right: quantification of nuclear/cytoplasmic FoxO1-GFP ratios at the end of day 14, $n = 3$ biologically independent samples. **c** Left: western blot of phospho-IRS1, phospho-AKT, phospho-FOXO and GAPDH protein abundance in *FTO<sup>rs9939609-TT</sup>*-myotubes: H1, H7, H9 and *FTO<sup>rs9939609-A</sup>*-myotubes: FTO-H1-1, FTO-H1-2, FTO-H7-1, FTO-H7-2, FTO-H9-1, FTO-H9-2 exposed to 1% HFD serum at day 14. Right: quantification of the abundance of insulin signaling proteins, $n = 3$ biologically independent samples. **d** Measurement of glucose uptake in *FTO<sup>rs9939609-TT</sup>*-myocytes and *FTO<sup>rs9939609-A</sup>*-myocytes in the presence of 1 µg/ml insulin or not (FTO-TT + 2-DG(-)/Insulin(-) vs. FTO-A + 2-DG(-)/Insulin(-), $P = 0.72198$; FTO-TT + 2-DG(+)/Insulin(-) vs. FTO-A + 2-

DG(+)/Insulin(-), $P = 0.23893$; FTO-TT + 2-DG(+)/Insulin(+) vs. FTO-A + 2-DG(+)/Insulin(+), $P = 0.03833$), $n = 3$ biologically independent samples. **e** Left: Western blot of phospho-IRS1, phospho-AKT, phospho-FOXO and GAPDH protein abundance in *FTO<sup>rs9939609-TT</sup>*-myotubes: H1, H7, H9 and *FTO<sup>rs9939609-A</sup>*-myotubes: FTO-H1-1, FTO-H1-2, FTO-H7-1, FTO-H7-2, FTO-H9-1, FTO-H9-2. Right: quantification of the abundance of insulin signaling proteins, $n = 3$ biologically independent samples. **f** Left: Western blot of phospho-AKT, phospho-FOXO and GAPDH protein abundance in *FTO<sup>rs9939609-TT</sup>*-myotubes: H1, H7, H9 and *FTO<sup>rs9939609-A</sup>*-myotubes: FTO-H1-1, FTO-H1-2, FTO-H7-1, FTO-H7-2, FTO-H9-1, FTO-H9-2, under serum-fed (Fed), 18 hr serum starved (Ss) or insulin-stimulated (Ins) conditions. Insulin stimulation was performed in serum-starved myotubes with 10 µg/mL insulin for 5 min. Right: quantification of the abundance of insulin signaling proteins, $n = 3$ biologically independent samples. H1 hESC line colored in purple, H7 hESC line colored in light blue and H9 hESC line colored in dark blue. Data are presented as mean + SEM. Data were analyzed using Wald test (**a**). $P$ values were calculated by two-tailed unpaired t-test. $*P < 0.05$, $**P < 0.01$. Source data are provided as a Source Data file.

levels (Supplementary Fig. 8a), and also downstream insulin/IGF signaling, *i.e.* the phosphorylation of IRS1, AKT, and FOXO. (Supplementary Fig. 8b). Similarly, inhibition with the FTO m<sup>6</sup>A demethylase-specific drugs bisantrene[45] and entacapone[46] showed that the FTO m<sup>6</sup>A demethylase activity is needed for the *FTO<sup>rs9939609-A</sup>* allele to increase *H19* and *IGF2* RNA levels (Fig. 5h). Although *H19* and IGF2 are known to be opposing actors in paternal/maternal imprinting at the epigenetic level, this mechanism does not apply in skeletal muscle tissues, because *H19* does not show any imprinting in human muscles. In fact, *H19* and IGF2 are complementary regulators in the skeletal muscles, via their synergistic effects on IGF signaling. It is known that increased *H19* lncRNA can sponge *let-7* microRNAs to enhance IGF signaling[44]. Thus, the *FTO<sup>rs9939609-A</sup>* allele enhances IGF signaling, and its downstream feedback responses, via increased FTO expression and thus enhanced m<sup>6</sup>A demethylation and stabilization of both *H19* and *IGF2* RNAs.

## Phenotypes induced by an edited *FTO* SNP can be blunted by FTO inhibition

To verify if our model is correct, we tested if low concentrations of FTO m<sup>6</sup>A demethylase inhibitors (bisantrene, entacapone) can specifically blunt the phenotypes of *FTO<sup>rs9939609-A</sup>* myoblasts and myotubes. First, we found that both FTO inhibitors could blunt the increased insulin/IGF sensitivity of *FTO<sup>rs9939609-A</sup>* myotubes under growth conditions; that is, before exposure to HFD serum, as determined by the levels of phospho-AKT and phospho-FOXO (Fig. 6a). Second, we found that one or both FTO inhibitors could weakly but significantly blunt the enhanced specification of *FTO<sup>rs9939609-A</sup>*, according to *PAX7, MYOD1, MYOG, ACTA1* and *MYHC* levels (Fig. 6b). Third, we found that both FTO inhibitors could also blunt the proliferative phenotype of *FTO<sup>rs9939609-A</sup>* myogenic progenitors (Fig. 6c). Fourth, we found that both FTO inhibitors could block the accelerated senescence of *FTO<sup>rs9939609-A</sup>* myogenic progenitors (Fig. 6d). Fifth, we found that both FTO inhibitors could normalize the insulin signaling and reverse the IR of *FTO<sup>rs9939609-A</sup>* myocytes, after 14-days exposure to HFD serum, as determined by the frequency of nuclear FoxO1-GFP translocation (Fig. 6e). Thus, the *FTO<sup>rs9939609-A</sup>* allele enhances insulin/IGF signaling, and its downstream paradoxical phenotypes, via increased FTO expression and enhanced m<sup>6</sup>A demethylation.

## Phenotypes induced by an edited *FTO* SNP can be blunted by IGF2 inhibition and *H19* knockdown

To further verify if our model is correct, we tested if low concentrations of an IGF2 monoclonal antibody can specifically blunt the phenotypes of *FTO<sup>rs9939609-A</sup>* myoblasts and myotubes. First, as expected of an IGF antagonist, we found that IGF2 antibody treatment could blunt the increased insulin/IGF signaling of *FTO<sup>rs9939609-A</sup>* myotubes displayed under growth conditions, as determined by the levels of phospho-AKT and phospho-FOXO (Fig. 7a). We also found

that IGF2 antibody could weakly but significantly blunt the enhanced specification of *FTO<sup>rs9939609-A</sup>*, according to *PAX3, PAX7, MYOD1* and *NCAM1* levels (Fig. 7b). We further found that IGF2 antibody treatment could also blunt the proliferative phenotype of *FTO<sup>rs9939609-A</sup>* myogenic progenitors (Fig. 7c), as well as the accelerated senescence of *FTO<sup>rs9939609-A</sup>* myogenic progenitors (Fig. 7d). Finally, we found that IGF2 antibody treatment could normalize the insulin signaling and reverse the IR of *FTO<sup>rs9939609-A</sup>* myocytes, after 14-day exposure to HFD serum, as determined by the frequency of nuclear FoxO1-GFP translocation (Fig. 7e).

Along these lines, we also found that knockdown of *H19* lncRNA with RNAi (Fig. 7f) significantly reversed the enhanced specification of *FTO<sup>rs9939609-A</sup>*, as determined by the protein levels of *PAX3, PAX7*, and *MYOD1* (Fig. 7g). Knockdown of *H19* also blunted the increased insulin signaling of *FTO<sup>rs9939609-A</sup>*, as determined by the levels of phosphorylated IRS1, FOXO, and AKT (Fig. 7g).

Altogether, our results confirmed that *FTO<sup>rs9939609-A</sup>* results in higher protein expression of FTO m<sup>6</sup>A demethylase during skeletal muscle development, thereby reducing *IGF2/H19* m<sup>6</sup>A and increasing *IGF2/H19* RNA expression to stimulate insulin/IGF signaling. Long-term over-stimulation of *IGF2/H19*-insulin signaling by *FTO<sup>rs9939609-A</sup>* in the context of replicative aging or obesogenic cues led to feedback inhibition of insulin/IGF signaling and thus senescence and muscle IR.

## Discussion

Obesity-associated FTO SNPs correlate with increased energy intake[47–53], and greater lean mass[37], but not physical activity levels[48,54–58]. These correlations are consistent with an increased BMI, but their cause-effect relationships with increased lean mass had remained unclear. In fact, the positive association of human *FTO<sup>rs9939609-A</sup>* with both fat mass and lean mass is paradoxical, because studies have indicated that lean mass is negatively correlated with obesity, IR and MetS[38]. Using hESCs to model multiple tissue developmental pathways, we found that *FTO<sup>rs9939609-A</sup>* had a surprisingly strong effect on muscle FTO and myogenesis compared to other tissue lineages. This is consistent with previous findings indicating (i) skeletal muscles are a major source of insulin/IGF-sensitivity[59], (ii) the presence of a nearby muscle-specific *FTO* enhancer that links to the *FTO* promoter[42], and (iii) the *FTO* SNPs' genetic effect on lean mass is so strong that it is unperturbed by environmental variables, such as dietary fat intake, unlike *FTO*'s association with fat mass[37]. Previous studies did find a positive link between FTO activity and the percentage of type I muscle fibers after mouse muscle regeneration, suggesting that FTO promotes type I myofiber formation in mice[60], which is consistent with our findings, though we found a more extensive promotion of all subtypes of myogenesis (Supplementary Fig. 4a) in human cells. According to a paper that examined GWAS for lean mass[33], both rs9936385 and rs11649091 were correlated with *FTO* expression in skeletal muscles,

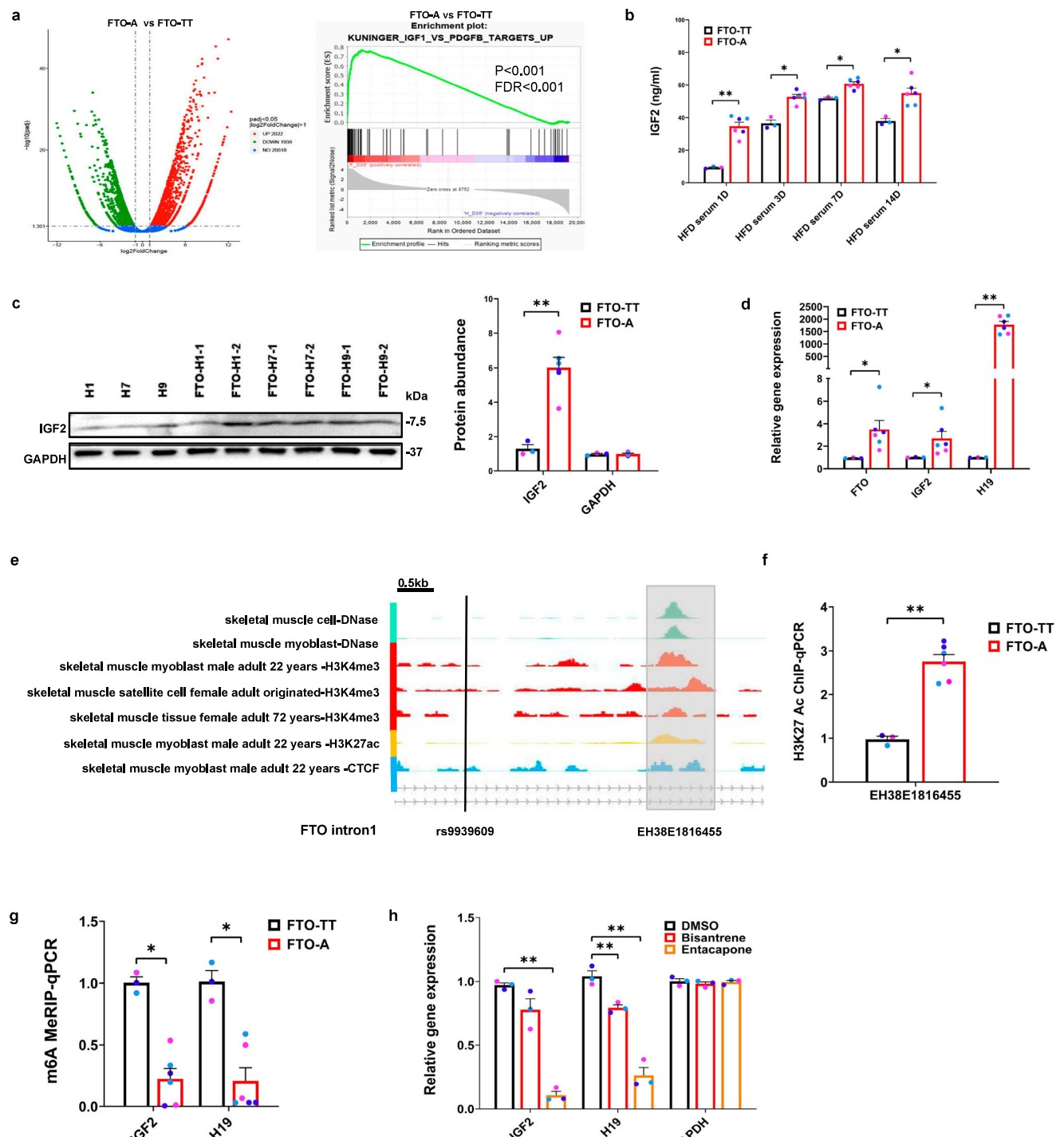

**Fig. 5 | *FTO^{rs9939609-A}* induces an increase in IGF2 and *H19* via m⁶A demethylation.**
**a** Left: Volcano plot of differentially expressed genes in *FTO^{rs9939609-A}*-myotubes, relative to *FTO^{rs9939609-TT}*-myotubes, according to RNA-sequencing. Right: Gene Set Enrichment Analysis (GSEA) of the IGF signature enriched in *FTO^{rs9939609-A}*-myotubes, relative to *FTO^{rs9939609-TT}*-myotubes. **b** ELISA of IGF2 protein secretion by *FTO^{rs9939609-TT}*-myotubes and *FTO^{rs9939609-A}*-myotubes into the culture medium after exposure to 1% HFD serum for 1 day, 3 days, 7 days, and 14 days, *n* = 3 biologically independent samples. **c** Left: Western blot of IGF2 and GAPDH protein abundance in *FTO^{rs9939609-TT}*-myotubes: H1, H7, H9 and *FTO^{rs9939609-A}*-myotubes: FTO-H1-1, FTO-H1-2, FTO-H7-1, FTO-H7-2, FTO-H9-1, FTO-H9-2. Right: quantification of the abundance of IGF2 and GAPDH proteins (FTO-TT vs. FTO-A, *P* = 0.00099), *n* = 3 biologically independent samples. **d** Quantitative RT-PCR for *FTO* mRNA, *IGF2* mRNA and *H19* lncRNA gene expression in *FTO^{rs9939609-TT}*-myotubes and

*FTO^{rs9939609-A}*-myotubes, *n* = 3 biologically independent samples. **e** Encode database: cis-Regulatory Elements near rs9939609 in skeletal muscle cells and skeletal muscle tissues. **f** H3K27Ac ChIP-qPCR of EH38E1816455 enhancer in *FTO^{rs9939609-TT}*-myotubes and *FTO^{rs9939609-A}*-myotubes (FTO-TT vs. FTO-A, *P* = 0.00019), *n* = 3 biologically independent samples. **g** N6-methyladenosine (m⁶A) MeRIP-qPCR for *IGF2* and *H19* in *FTO^{rs9939609-TT}*-myotubes and *FTO^{rs9939609-A}*-myotubes, *n* = 3 biologically independent samples. **h** Quantitative RT-PCR for *IGF2* and *H19* lncRNA gene expression in *FTO^{rs9939609-TT}*-myotubes, treated with DMSO, 200 nM bisantrene or 50 μM entacapone, *n* = 3 biologically independent samples. H1 hESC line colored in purple, H7 hESC line colored in light blue and H9 hESC line colored in dark blue. Data are presented as mean + SEM. *P* values were calculated by two-tailed unpaired t-test. *P < 0.05, **P < 0.01. Source data are provided as a Source Data file.

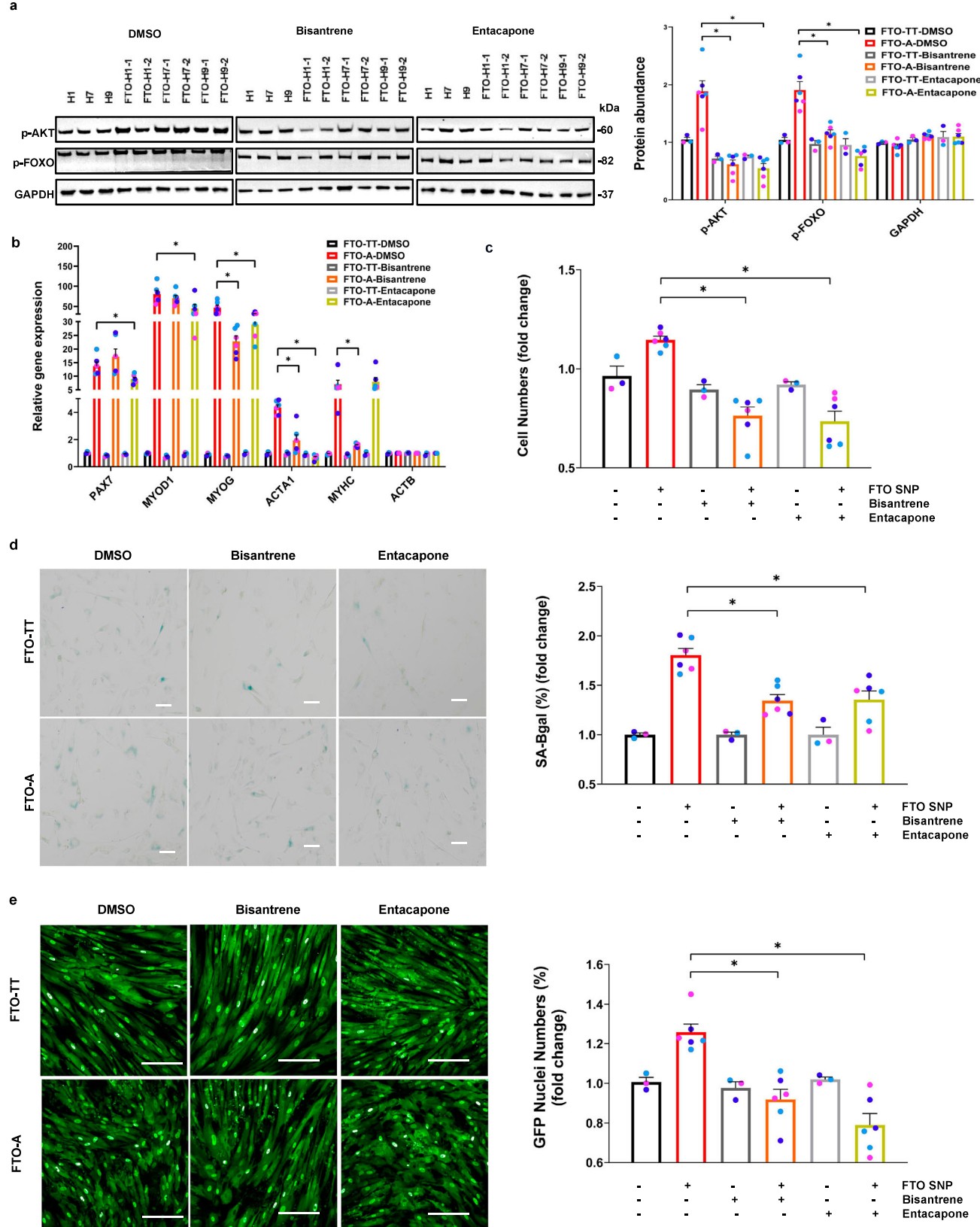

but not in the adipose or liver tissue. Moreover, in the GTEx database, there were clearly no eQTLs near FTOrs9939609 in the heart nor the brain, in contrast to skeletal muscle tissue (Supplementary Figs. 6b, c). We have further shown that the *FTO$^{rs9939609-A}$* SNP leads to *IGF2* and *H19* overexpression, thus increasing insulin/IGF signaling and muscle growth during development. Given the inherent differences between

human and mouse gene networks, including evolutionary differences in *FTO* and *H19/IGF2* expression[61–63] (e.g., in humans, plasma IGF2 is high throughout life, but drops to near-zero in adult mice except during regeneration), mouse *H19* and *Igf2* are likely not regulated by human FTO. In the clinic, *FTO*-associated overgrowth is reminiscent of the Beckwith-Wiedemann syndrome (BWS), which leads to overgrowth

**Fig. 6 | Phenotypes induced by *FTO^rs9939609-A* are reversed by FTO inhibition.**
**a** Left: Western blot of phospho-AKT, phospho-FOXO and GAPDH protein abundance in *FTO^rs9939609-TT*-myotubes: H1, H7, H9 and *FTO^rs9939609-A*-myotubes: FTO-H1-1, FTO-H1-2, FTO-H7-1, FTO-H7-2, FTO-H9-1, FTO-H9-2, treated with DMSO, 200 nM bisantrene or 50 μM entacapone for 7 days. Right: quantification of the abundance of insulin signaling proteins, *n* = 3 biologically independent samples. **b** Quantitative RT-PCR for myogenic markers in *FTO^rs9939609-TT*-myotubes and *FTO^rs9939609-A*-myotubes treated with DMSO, 200 nM bisantrene or 50 μM entacapone for 7 days, *n* = 3 biologically independent samples. **c** Quantification of fold change in *FTO^rs9939609-TT*-myoblasts and *FTO^rs9939609-A*-myoblasts numbers, treated with DMSO, 200 nM bisantrene or 50 μM entacapone for 7 days, *n* = 3 biologically independent samples. **d** Left: representative images of senescent *FTO^rs9939609-TT*-myoblasts and *FTO^rs9939609-A*-myoblasts, treated with DMSO, 200 nM bisantrene or 50 μM entacapone FTO inhibition for 7 days by passage 16. Senescent cells were stained for β-galactosidase (SA-βgal). Scale bars, 100 μm. Right: quantification of percentage of SA-βgal+ senescent *FTO^rs9939609-TT*-myoblasts and *FTO^rs9939609A*-myoblasts, *n* = 3 biologically independent samples. **e** Left: representative images of FoxO1-GFP+ nuclei numbers at the end of 14 days' exposure to 1% HFD serum, treated with DMSO, 200 nM bisantrene or 50 μM entacapone. Scale bars, 200 μm. Right: quantification of the percentage of FoxO1-GFP+ nuclei numbers, *n* = 3 biologically independent samples. H1 hESC line colored in purple, H7 hESC line colored in light blue and H9 hESC line colored in dark blue. Data are presented as mean + SEM. *P* values were calculated by two-tailed unpaired t-test. *P < 0.05, **P < 0.01. Source data are provided as a Source Data file.

and IR, which also implicates both *IGF2* and *H19*, and which is imperfectly modeled by any single mouse model[64]: BWS is a human genomic imprinting disorder at a locus harboring *IGF2* and *H19*, that is characterized by overgrowth and early onset obesity[65], although BWS is often an underestimated cause of syndromic obesity. Like BWS, the *FTO* polymorphisms also often lead to early onset obesity in children and adolescents[66]. It is likely that through a process of natural selection, genetic traits like *FTO^rs9939609-A* that promote overgrowth have become prevalent due to the evolutionary advantages they offered in the wild[67].

A curious phenomenon that we observed with insulin signaling in *FTO^rs9939609-A* myotubes is that the SNP can both promote and suppress insulin/IGF-PI3K-AKT signaling. Such opposing effects have been documented for other cytokines as well. A widespread feature of extracellular signaling in cellular gene circuits is paradoxical pleiotropy. In such paradoxical signaling, the same secreted signaling molecule can induce opposite effects in the responding cells[68]. For example, glucose has both mitogenic and toxic effects on pancreatic β cells[69]. In developing fly embryos, an inhibitor of BMP also increases BMP's range of activity[70,71]. In the immune system, the same cytokines can both promote and also inhibit T cell activities[72-75]. In the skeletal muscles, the pro-inflammatory IL-6 cytokine is known to promote insulin sensitivity as a myokine after exercise, but a high-level increase due to chronic disease leads to insulin resistance[62]. Here, we are observing a similar effect, where (FTO-induced) *H19* and *IGF2* both promote insulin signaling in the short-term, and suppress insulin signaling in the long-term. Synthetic biology experiments with genetic circuits have shown that circuits with such paradoxical signaling are more robust to perturbations to maintain optimal signaling[76,77]. FTO-m6A-*H19/IGF2*-insulin signaling may have evolved to maintain skeletal muscle cells within an optimal range of growth rates. Unfortunately, this circuit has become maladaptive in modern environments. In fact, obesogenic *FTO* alleles might accelerate both developmental growth and ageing[78]. Both insulin and IGFs are able to bind to and activate each other's receptors, albeit with reduced affinity, and both receptors also elicit common downstream kinase signaling[79]. Excessive insulin/IGF signaling could be the basis for IR[80,81], where muscles and adipose tissues simultaneously develop a vicious cycle of excessive fatty acid metabolism and inflammation, thus causing IR and further worsening of hyperlipidemia, ultimately leading to reduced glucose uptake and MetS during aging[82]. It has been long-debated whether the original source of IR that eventually leads to MetS and T2D are skeletal muscles[83,84] or adipose tissue or liver hepatocytes[85]. It may ultimately depend on each individual's lifestyle and complex genotype, but our data here suggests that at least for individuals who carry high-risk *FTO* SNPs, their dietary and drug regimens should be tailored to prevent muscle IR.

As *FTO^rs9939609-A* myocytes showed accelerated development, growth and IR, they also underwent aging prematurely and became senescent. Such acceleration of developmental timing and aging is consistent with associations between *FTO* and human developmental traits, such as accelerated childhood obesity[78], the age of puberty/menarche of adult women[86], and that both the *FTO^rs9939609-A* SNP and excessively high IGF levels are known to increase mortality risk and affect aging[37,40,87–96]. In fact, IGF2 is also associated with aging-related stem cell exhaustion[97]. One other human overgrowth syndrome that eventually leads to IR, T2D and higher mortality rates, which also implicates excessive insulin/IGF signaling and insulin/IGF resistance, is acromegaly[40,98]. Taken together, our results on muscle FTO-*H19/IGF2*-insulin signaling explain how the *FTO^rs9939609-A* SNP might promote aging. Moreover, our findings can explain some of the confusing human genetics findings on related *FTO* SNPs, including with regards to sarcopenia. Although several GWASs have shown that *FTO* SNPs positively associate with both lean mass and fat mass as a "sumo wrestler" gene locus[33,37,99], another recent study showed that *FTO^rs9939609-A* is associated with over 3-fold higher risk of sarcopenia[100]. This paradox can be easily understood and resolved based on our *H19*-IGF2 findings, as young adults display the early phase of growth due to FTO-m6A-*H19*-IGF2-insulin signaling, whereas the elderly manifest with sarcopenia in the later phase due to excessive insulin signaling and insulin resistance. More generally, our results showcase the potential of using this CRISPR-hESC-tissue platform to model and resolve other outstanding problems in the genetics of human growth and metabolism. It is no coincidence that most growth-associated SNPs are almost exclusively enriched with muscle DNAse sites[101]. Already we were able to show that an edited *FTO^rs9939609-A* SNP can lead to higher expression of muscle *FTO* (but not muscle *IRX3* nor *IRX5* nor *RPGRIP1L*, likely due to different SNP-enhancer mechanisms being active in different cell-types[22,23]) in human muscle cells, and discover that FTO-m6A-*H19*-IGF2-insulin signaling can simultaneously explain both the paradoxical (pro-growth and pro-IR) phenotypes of *FTO^rs9939609-A* and that only HFD serum (but not oft-used IR inducers) can cooperate with the *FTO^rs9939609-A* genotype to induce muscle IR, leading to a new physiological model of human muscle IR. It has been long-debated whether skeletal muscles are the initial tissue that displays IR that eventually leads to MetS and T2D[84,85]. Warram et al. indicate that one to two decades before type II diabetes is diagnosed, reduced glucose clearance is already present. This reduced clearance is accompanied by compensatory hyperinsulinemia, not hypoinsulinemia, suggesting that the primary defect is in peripheral tissue response to insulin and glucose, not in the pancreatic beta cell[102]. Our results suggest that, at least in carriers with the *FTO^rs9939609-A* SNP, skeletal muscles are more easily susceptible to excessive insulin/IGF signaling and IR than other tissues, during aging and exposure to high-fat diets. Our results also support a model where muscle *FTO^rs9939609-A* directly promotes muscle IR and indirectly promotes adipose and liver IR in tandem in a vicious cycle during sarcopenia and obesity. In the future, the CRISPR-hESC-tissue platform approach could be further harnessed to study more of such genotype-phenotype questions.

## Methods
### Mice
High fat diet (HFD) fed mouse (male, 19-week-old) were purchased from Vital River (Beijing, China). To generate DIO models, HF diet (60% kcal/

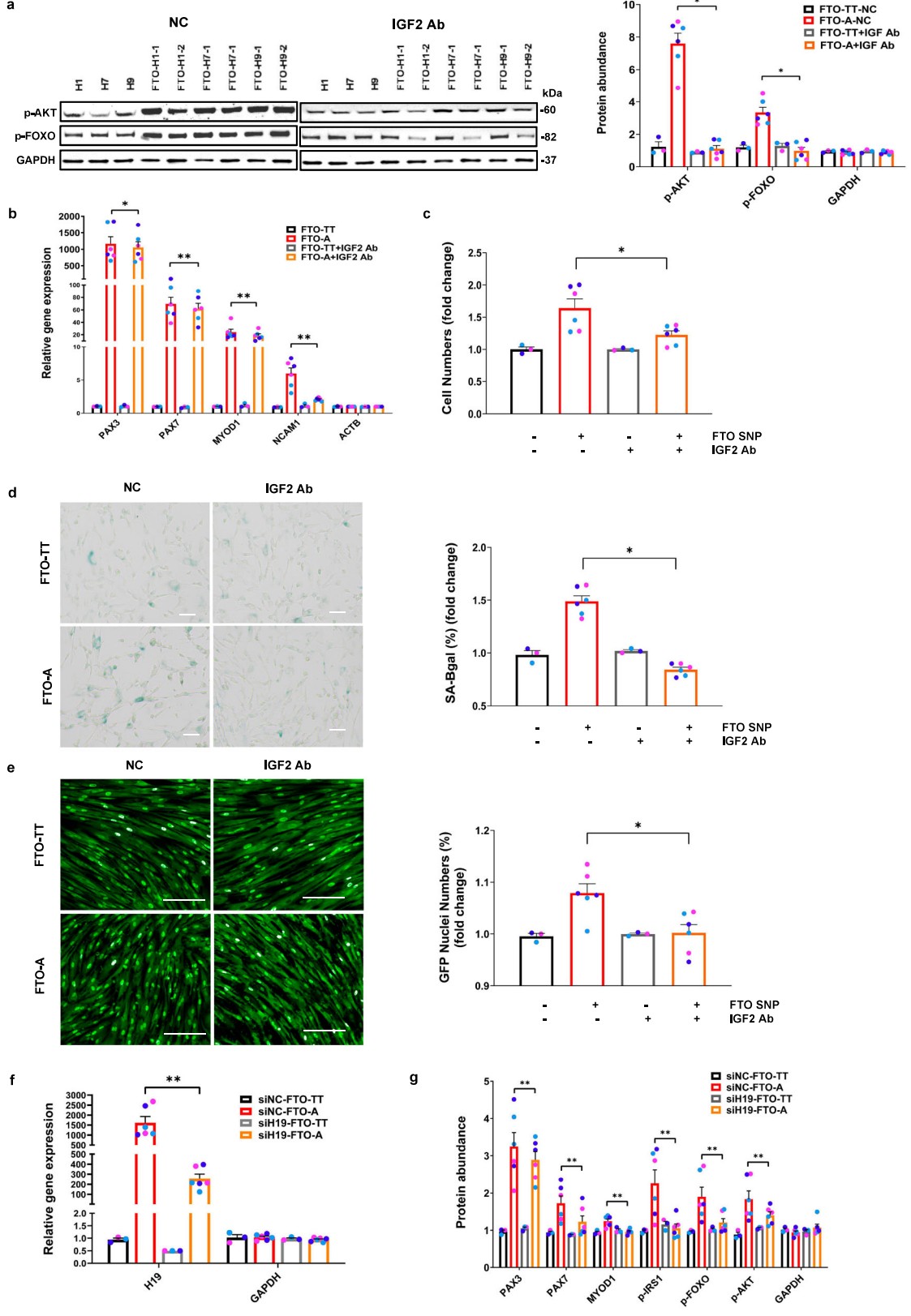

fat, D12492, Research Diets) were introduced at 6-weeks and fed for 13 consecutive weeks before purchase. All mice were housed in a pathogen-free room conditioned at 20-22 °C with alternating 12 h cycles of light and dark, and with free access to pellet food and water. Blood was centrifuged at 2000 g for 15 min. Serum was then separated and stored at −80 °C. All animal experiments and procedures were approved by the Institutional Animal Care and Use Committee (IACUC) of the Institute of Zoology (Chinese Academy of Sciences, IOZ-IACUC-2022-170).

## hESC cultures

H1, H7 and H9 hESCs (WiCell) were independently derived from different human blastocysts and well-established as pluripotent stem cell

**Fig. 7 | Phenotypes induced by *FTO*^rs9939609-A* are reversed by IGF2 inhibition and *H19* knockdown. a** Left: Western blot of phospho-AKT, phospho-FOXO and GAPDH protein abundance in *FTO*^rs9939609-TT*-myotubes: H1, H7, H9 and *FTO*^rs9939609-A*-myotubes: FTO-H1-1, FTO-H1-2, FTO-H7-1, FTO-H7-2, FTO-H9-1, FTO-H9-2, with or without 0.8 µg/ml human IGF2 antibody inhibition for 7 days. Right: quantification of the abundance of insulin signaling proteins, *n* = 3 biologically independent samples. **b** Quantitative RT-PCR for myogenic markers in *FTO*^rs9939609-TT*-myotubes and *FTO*^rs9939609-A*-myotubes treated with or without 0.8 µg/ml human IGF2 antibody inhibition for 7 days, *n* = 3 biologically independent samples. **c** Quantification of fold change in *FTO*^rs9939609-TT*-myoblasts and *FTO*^rs9939609-A*-myoblasts numbers, with or without 0.8 µg/ml human IGF2 antibody inhibition for 7 days (FTO-TT vs. FTO-A, *P* = 0.02165), *n* = 3 biologically independent samples. **d** Left: representative images of senescent *FTO*^rs9939609-TT*-myoblasts and *FTO*^rs9939609-A*-myoblasts, with or without human IGF2 antibody inhibition for 7 days by passage 16. Senescent cells were stained for β-galactosidase (SA-βgal). Scale bars, 100 µm. Right: quantification of percentage of SA-βgal+ senescent *FTO*^rs9939609-TT*-myoblasts and *FTO*^rs9939609-A*-myoblasts, *n* = 3 biologically independent samples. **e** Left: representative images of FoxO1-GFP+ nuclei numbers at the end of 14 days' exposure to 1% HFD serum, with or without human IGF2 antibody inhibition. Scale bars, 200 µm. Right: quantification of the percentage of FoxO1-GFP+ nuclei numbers (FTO-TT vs. FTO-A, *P* = 0.01002), *n* = 3 biologically independent samples. **f** Quantitative RT-PCR for *H19* and *GAPDH* (FTO-TT vs. FTO-A, *P* = 0.00136), *n* = 3 biologically independent samples. siNC: knockdown negative control; siH19: knockdown of *H19* lncRNA. **g** Quantification of the abundance of myogenic and insulin signaling proteins, *n* = 3 biologically independent samples. siNC: knockdown negative control; siH19: knockdown of *H19* lncRNA. H1 hESC line colored in purple, H7 hESC line colored in light blue and H9 hESC line colored in dark blue. Data are presented as mean + SEM. *P* values were calculated by two-tailed unpaired *t*-test. *\*P* < 0.05, *\*\*P* < 0.01. Source data are provided as a Source Data file.

lines[103–105]. All hESCs were tested to be mycoplasma-free and cultured feeder-free in mTeSR1 media (StemCell Technologies) on cell-culture plates that had been pre-coated with Matrigel (Corning) basement membrane matrices at 37 °C and 5% CO$_2$. Each day, mTeSR1 media was changed and hESC cultures were visually inspected with care to avoid any spontaneous differentiation. When partially confluent, hESCs were serially passaged as small clumps by removing mTeSR1, and then adding an EDTA solution (Gibco) for 5 min at 37 °C to partially dissociate the monolayers. Subsequently, EDTA was removed, fresh mTeSR1 was added, and then hESCs were scraped off the plate using a cell scraper and then transferred to new Matrigel-coated plates in mTeSR1 media.

### Genome editing of hESCs

To genetically engineer *FTO*^rs9939609-TT* H1, H7 and H9 hESCs to carry the *FTO*^rs9939609-A* allele at the endogenous *FTO* locus, a point mutation was introduced into the endogenous *FTO* locus using CRISPR-Cas9 prime editing technology. 500,000 hESCs were mixed with the nucleofection solution containing Cas9, RT and pegRNAs. The hESCs were then electroporated in a 16-well Nucleocuvette Strip, using the 4D Nucleofector system. Following electroporation, cells were plated into one well of a Matrigel-coated 24-well plate containing 500 µL of mTeSR1 media supplemented with 10 µM ROCK inhibitor (Y27632). After 48 h, mTeSR1 alone was used (without Y27632). After CRISPR prime editing, hESC single cells were expanded as clonal lines for 9 passages, then screened via genomic sequencing to confirm successful knock-in. The pegRNAs and primers used are listed in Supplementary Data 1. Six *FTO*^rs9939609-A* hESC lines and their isogenic *FTO*^rs9939609-TT* hESC control lines, all cloned, expanded and cultured in parallel, were used in all profiling and perturbation experiments.

### Differentiation into ectodermal, endodermal and mesodermal progenitors for hESC-tissue models and organoids

For directed differentiation into neuroectodermal (neural) progenitors, endodermal (liver) progenitors, and mesodermal (adipocyte, cardiac muscle, skeletal muscle) progenitors, we followed previously established protocols[39,106–109] (reagents listed in Supplementary Data 2). Multiple protocols were attempted for mesoderm differentiation[110–114], and the best protocol in terms of myoblast percentage was selected. All culture media were refreshed daily throughout the protocols. Human myoblasts were expanded in DMEM/F12 expansion medium supplemented with 20% FBS (BioIND) and 1% penicillin–streptomycin (Gibco) and passaged at less than 80% confluence using Trypsin-EDTA (Gibco). To differentiate into myotubes, myoblasts were seeded at 80% confluency and differentiation medium supplemented with 2% B27 (Gibco) was added the next day. Multinucleated mature myotubes formed within 7 days of adding differentiation medium.

### RNA purification and Reverse transcription

Total RNA was purified with TRIzol reagents (Thermo Scientific) and was reverse transcribed into cDNA using the PrimeScript RT reagent Kit with gDNA Eraser (Perfect Real Time) (Takara). RNA concentration was measured by NanoDrop 8000 Spectrophotometer (Thermo Scientific) or a Qubit Fluorometer (Thermo Scientific).

### RT-qPCR and MeRIP-RT-qPCR

RT-qPCR was conducted by using SYBR FAST qPCR Kits (KAPA Biosystem) according to the manufacturer's protocol and was performed using a LightCycler480 II detection system (Roche Applied Science). Relative expression level of each gene was normalized to the reference gene ACTB or GAPDH.

MeRIP to enrich for m6A-modified RNA with immunoprecipitation, without fragmentation of total RNA, was performed using the m6A antibody (ab151230, Abcam)[115] Non-m6A input RNA was used as the normalization controls for m6A analysis in MeRIP-RT-qPCR. Primers used for qPCR are listed in Supplementary Data 1.

### Western blots

Protein extracts were prepared with RIPA lysis and extraction buffer (Thermo Scientific) with protease and phosphatase inhibitor cocktail (Thermo Scientific). All lysates were quantified by BCA protein assay (Thermo Scientific). Ten micrograms of protein from each sample were electrophoresed on 12% sodium dodecyl sulfate-polyacrylamide gel electrophoresis and transferred to nitrocellulose membranes (Merck Millipore). Blots were incubated with primary antibodies: FTO (ab94482; 1:1000; Abcam), PAX3 (AB528426; 0.5 µg/ml; DSHB), PAX7 (AB528428; 0.5 µg/ml; DSHB), MYOD1 (sc-377460; 1:200; Santa Cruz), MYOG (sc-52903; 1:200; Santa Cruz), MHC/MF20 (AB2147781; 0.5 µg/ml; DSHB), p-IRS1 (2385S; 1:1000; Cell Signaling), p-FOXO1 (9461S; 1:1000; Cell Signaling), p-AKT (4060S; 1:1000; Cell Signaling), IGF2 (ab9574; 1:700; Abcam) and GAPDH (2118S; 1:1000; Cell Signaling), then probed with the secondary antibody anti-mouse IgG HRP (7076S; 1:1000; Cell Signaling) or anti-rabbit IgG HRP (7074S; 1:1000; Cell Signaling). HRP-based detection was performed using an iBright FL1000 Imaging System (Invitrogen).

### Sequencing

Total DNA and RNA were extracted from cells and quantitated using a Qubit Fluorometer, and the integrity was assessed using the Bioanalyzer 2100 system (Agilent Technologies, CA, USA). DNA-sequencing libraries were generated using the TruSeq Library Construction Kit (Illumina) following manufacturer's recommendations, and index codes were added to each sample. In brief, genomic DNA was fragmented by sonication to a median size of 350 bp. Then, DNA fragments were end-repaired, A-tailed, and ligated with the full-length Illumina sequencing adapters, followed by further PCR amplification. RNA-sequencing libraries were generated using NEBNext® UltraTM RNA Library Prep Kit for Illumina® (NEB, USA) following manufacturer's recommendations and index codes were added to attribute sequences to each sample. Libraries were analyzed for size distribution using an Agilent Bioanalyzer 2100 and quantified via real-time PCR. Sequencing

was performed on an Illumina Novaseq platform and 150 bp paired-end reads were generated. Raw data (raw reads) of fastq format were firstly processed through in-house perl scripts. In this step, clean data (clean reads) were obtained by removing reads containing adapter, reads containing poly-N and low quality reads from raw data. At the same time, Q20, Q30 and GC content the clean data were calculated. DNA-sequencing reads were aligned to the Human Genome Reference Consortium build 37 (GRCh37) using BWA v.0.7.8 (BWA-mem). Unaligned reads that passed Illumina's quality filter (PF reads) were retained. The average raw data of each sample was 68.63 G, the average effective data was 98.92%, the average Q20 was 97.22% and the average Q30 was 92.29%. In conclusion, the sequencing data quality met the analysis requirements. The average sequencing depth of each sample was no less than 20X and the coverage was no less than 99.85%. RNA-sequencing reads were aligned to the Ensembl reference genome and gene model annotation files were downloaded from the Ensembl website directly. The average clean data of each sample was no less than 10 G, and the total mapping was no less than 97%. Differential expression analysis of two conditions was performed using the edgeR R package. The $P$ values were adjusted using the Benjamini & Hochberg method. Corrected $P$-value of 0.05 and absolute fold change of 2 were set as the threshold for significantly differential expression.

## Myotube fusion index

Fusion index was assessed by calculating the ratio of multinucleated myotube nuclei to the total number of nuclei. Myotubes were washed with PBS and fixed with 2% paraformaldehyde. Myotubes were incubated with anti-MHC/MF20 (DSHB, 2.5 μg/ml), followed by secondary antibody anti-mouse IgG Cy3 (1:500; Thermo Scientific). DAPI (10 μg/m; Beyotime) staining was used for nuclei counting. Stained cells were automatically imaged with a PerkinElmer Operetta CLS, and at least 6 images per group were included in the statistics.

## Myotubes area, width and length

On day 7 after differentiation, myoblasts were differentiated into myotubes. Then myotubes were washed with PBS and fixed with 2% paraformaldehyde and were incubated with anti-MHC/MF20, followed by secondary antibody anti-mouse IgG Cy3 (1:500; Thermo Scientific). Images were acquired with a PerkinElmer Operetta CLS. Images were automatically analyzed by using Harmony 4.6 software. Whole myotubes were defined using cell area, width and length, after recognition with linear classifier-based machine learning (ML) algorithms trained and optimized in-house.

## Contractile myotubes

After 14 days of directed differentiation, human myotubes matured into contractile myofibers. Videos of the contractile myotubes were captured on a Nikon fluorescence microscope, and the percentage of contractile myotube areas were calculated.

## PDMS molds and 3D myobundles formation

6-well plates were poured with 3 mL of liquid polydimethylsiloxane (PDMS) 184 Silicone Elastomer Kit (10:1, Dow Corning Corps., Michigan, USA). The PDMS was cured by incubating the plates at 37 °C for 8–12 h. Two 27 G metal needles (as skeleton) were inserted into each well of PDMS molds with horizontal symmetry, and one end of two surgical silk (1 cm, 1–0 diameter) threads were held in place by two metal needles, respectively. In particular, two silk threads (as tendons) were kept in the same horizontal direction. Each well was soaked and sterilized with 70% ethanol at room temperature in a sterile sealed box for 0.5–3 h, then PDMS plates were further dried and sterilized by ultraviolet irradiation for 8–12 h after three washes with PBS supplemented with 2% penicillin-streptomycin[116].

PDMS molds were coated with 0.2% (w/v) pluronic F127 (Thermofisher) to prevent gel adhesion and incubated for 30 minutes at 37 °C. Pluronic F127 was aspirated and molds were washed with PBS before seeding human muscle cells. $5 \times 10^6$ human myoblasts/gel mixture consisted of 400 μL 2%B27 medium, 200 μLl 20 mg/mL fibrinogen (Sigma), 40 μL 250 U/mL thrombin (Sigma) and 1 mg/mL 6-aminocaproic acid (6-AA, Sigma) for each 6-well PDMS molds. Cells/gel mixture was injected into the PDMS molds and polymerized at 15 minutes at room temperature and cured at 37 °C and 5%CO2 for 0.5 h subsequently. After 24 h, growth medium supplemented with 1 mg/mL 6-AA for high-density monolayers culture was switched to a differentiation medium 2%B27 medium for 10-day differentiation, supplemented with 1 mg/mL 6-AA to prevent fibrin degradation.

## Cell proliferation assay

At day 0, $1.5 \times 10^4$ human myoblasts/well were seeded in a 24-well plate and maintained in myoblast expansion medium. Nuclear images were analyzed for Hoechst 33342 staining at days 0,1, 2 and 3. Cell numbers were counted using the Harmony 4.6 software automatically.

## Senescence-associated β-galactosidase assay

Human myoblasts were cultured on 24-well plates, and cultured to a cell density of 50% using expansion medium. Cells were fixed with glutaraldehyde and washed gently with 1 mL of DPBS prior to the assay. Senescence-associated β-galactosidase staining was performed with a kit according to the manufacturer's instructions (Beyotime). Three optical fields were randomly selected and imaged under an inverted microscope (Nikon) to calculate the number of senescent cells.

## Insulin resistance model

Human myoblasts were infected with lentiviral FoxO1-GFP for stable expression. Lentiviral plasmids of pLenti-FoxO1-Clover (Addgene #67759) were co-transfected with packaging vectors into HEK293FT cells. Supernatants were harvested after 48 h. Human myoblasts were infected with the lentiviral supernatants, containing 8 mg/mL polybrene, to achieve stable expression.

Whole blood was collected from HFD mice and serum was isolated using a serum separator tube (SST). Samples were allowed to clot for two hours at room temperature or overnight at 4 °C before centrifugation for 15 minutes at $1000 \times g$. Samples were aliquoted and stored at −80 °C without freeze-thaw cycles. Human myocytes were cultured in a differentiation medium supplemented with 1% HFD mouse serum for 14 days and different titrations of IR inducers, such as palmitate, ceramide, IL-1b, IL-6, and TNFa, and their combinations. Nucleo-cytoplasmic translocation of FoxO1-GFP fluorescence was analyzed using the PerkinElmer Operetta CLS, at days 0 and 14.

## Serum starvation

Western blot analysis of human myotubes, under serum-fed (Fed), 18 hr serum starved (Ss) or insulin-stimulated (Ins) conditions. Insulin stimulation was performed in serum-starved myotubes with 10 μg/mL insulin for 5 min.

## Screen quest™ colorimetric glucose uptake assay

Cells were plated in growth medium at 80,000 cells/well/100 μL in 96-well black wall/clear bottom cell culture plates for 4–6 h before experiments. After aspirating the medium from the wells, and incubating the cells with 100 μL/well of serum free medium, the cells were incubated at 37 °C, 5% $CO_2$ incubator overnight. Glucose uptake was then quantified using a Screen Quest™ Colorimetric Glucose Uptake Assay Kit (AAT Bioquest) according to the manufacturer's instructions.

## ELISA

Centrifuge samples for 20 minutes at $1000 \times g$. Collect the supernates and assay immediately. IGF1 and IGF2 levels of human myotubes were quantified using an ELISA kit according to the manufacturer's instructions.

## ChIP-RT-qPCR

For ChIP--RT-qPCR, samples were fixed with formaldehyde, lysed, sonicated, and precleared. The chromatin was probed overnight using H3K27Ac antibody (CST #4353) conjugated to Protein G Dynabeads (Invitrogen). Subsequently, chromatin was precipitated, rigorously washed, and de-crosslinked at 65 °C. Input DNA was used as the normalization controls for H3K27ac analysis in ChIP-RT-qPCR. Primers used for qPCR are listed in Supplementary Data 1.

## Inhibition of IGF2

Human myoblasts were treated with 0.8 μg/ml of Human IGF-II antibody (AF-292-NA, RnD) in expansion medium for 3 days for the cell proliferation, senescence-associated β-galactosidase staining assay and differentiation medium for 7 days for the RT-PCR. Insulin resistant myotubes were treated with 0.8 μg/ml Human IGF-II antibody in DMEM medium supplemented with 1% HFD mouse serum for 14 days, and the percentage of FoxO1-GFP+ nuclei was analyzed using the PerkinElmer Operetta CLS, at days 0 and 14.

## Inhibition of FTO

Human myoblasts were treated with DMSO, 200 nM Bisantrene or 50 μM Entacapone in expansion medium for 3 days for the cell proliferation assay, senescence-associated β-galactosidase staining assay and differentiation medium for 7 days for the RT-PCR. Insulin resistant myotubes were treated with DMSO, 200 nM Bisantrene or 50 μM Entacapone in DMEM medium supplemented with 1% HFD mouse serum for 7 days, and the percentage of FoxO1-GFP+ nuclei was analyzed using the PerkinElmer Operetta CLS, at days 0 and 7.

## Loss of function of *H19* and *FTO*

*H19* lncRNA and *FTO* was knocked down by siRNA. *H19*-specific siRNA, *FTO*-specific siRNA and negative controls were synthesized by Tsingke (Beijing, China). RNAi negative control was used as a negative control (NC). The siRNA sequences are listed in Supplementary Data 1. Transfection was achieved using Lipofectamine 3000 Transfection Reagent (Invitrogen) following the manufacturer's instructions.

## Statistical analysis

All data are represented as mean value + SEM for statistical comparisons. Statistical analysis was performed by using GraphPad Prism 10.0 and Microsoft Excel. Statistical analyzes were performed using Student's unpaired two-tailed *t*-tests. *P*-values less than 0.05 were considered statistically significant (*$P < 0.05$, **$P < 0.01$). Data for which a specific *P* value is not indicated are not significantly different.

## Reporting summary

Further information on research design is available in the Nature Portfolio Reporting Summary linked to this article.

# Data availability

All data are available in the main text or the supplementary materials. The sequencing data generated in this study have been deposited in the NCBI's Gene Expression Omnibus data bank under accession code GSE262261. Any additional information is available upon request to the corresponding author (Ng Shyh-Chang, huangsq@ioz.ac.cn). Source data are provided with this paper.

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

## Acknowledgements

We thank the State Key Laboratory of Stem Cell and Reproductive Biology and all members of the SRLab/KLORR for their kind support. This work was supported by the National Key R&D Program of China (2019YFA0801700), the National Natural Science Foundation of China (U21A20396), the Strategic Priority Research Program of the CAS (XDA16010109), the CAS Project for Young Scientists in Basic Research (YSBR-012), and the Howard Hughes Medical Institute (HHMI) International Scholar award.

## Author contributions

L.G., S.M., Z.Y., D.S., Y.C., S.L., P.W., J.S., Y.W., L.L. designed and performed the experiments. L.G. and N.S.-C. wrote the manuscript. N.S.-C. designed and supervised the entire project. All authors have given approval to the final version of the manuscript.

## Competing interests

The authors declare no competing interests.
