## [Peer Review File · Nature Communications]

REVIEWER COMMENTS

Reviewer #1 (Remarks to the Author):

This is an interesting report addressing a pertinent research questions with novel observations that could transform our understanding of the role of FTO. There is an unmet need for insight into the mechanisms that regulate sarcopenia and this paper adds to our understanding. However, some concerns need to be addressed:

Skeletal muscle growth is reliant on a harmonious proliferation of diverse cell types, not solely the myogenic progenitors explored in this manuscript. Data from the Rando lab has underscored the pivotal role of mesenchymal progenitors in muscle growth (PMID: 31091443). How does this information align with the relatively modest effects observed in adipose progenitors? What impact does FTOs9939609-A have on PDGFRalpha positive cells? The isolated impact on myogenic progenitors might suggest a crucial function in recovering from muscle injuries, potentially providing defense against sarcopenia. What existing knowledge exists regarding recovery from muscle injuries in FTO transgenic models?

The examinations into insulin resistance reveal a correlation between FTOs9939609-A and insulin signaling. Nevertheless, the insulin doses utilized could potentially activate the IGF1 receptor as well, introducing complexities into interpreting PI3k/Akt signaling. This should be acknowledged in the discussion. Numerous data sets have been unable to establish links between insulin signaling activity and glucose uptake. Should assertions be made regarding heightened insulin sensitivity, it will be imperative to observe modifications in insulin-triggered substrate metabolism.

Reviewer #2 (Remarks to the Author):

Guang et al used CRISPR prime editing to knock-in the FTOs9939609-A SNP into isogenic hESCs, which were then differentiated into tissue models, to study potentially hidden roles of the edited FTO SNP during human tissue development. They found that among the 6 lineages tested, FTOs9939609-A had an effect on the human skeletal muscle lineage where they report effects of muscle progenitor proliferation, differentiation, senescence and muscle insulin resistance.

I have some methodological queries and concerns which should be addressed to improve the quality and relevance of this work:

- The specific protocol used for myogenic differentiation is not very clear and only cross-referenced via a previous paper for the same group. Overall it also appears to have a limited efficiency in terms of differentiation and an unclear purity on the output (no purification steps). It is therefore critical to validate the findings across multiple (e.g. 3) different protocols to differentiate hPSCs into muscle to make sure the effect is not protocol specific. As an example, Figure 1b: these cells are supposed to be myotubes but they have remarkably low levels of MyHC (MF20) and high levels of Pax3/7. This profile is more of late myoblasts.
- Please show the individual data of each clone (i.e. for H1, H7 and H9 hESCs) across all experiments to better quantify the variability across the different lines
- I assume the controls here have also been subjected to the same clonal procedure that the edited cells have undergone?

Reviewer #3 (Remarks to the Author):

In this paper the authors use human embryonic stem cells in which they perform prime editing of the SNP rs9939609 that associates with obesity but also with lean mass. The stem cells are then differentiated to six different cell types, and a phenotypic effect of the allele editing is observed in skeletal muscle-like cells, in which functional assays are performed to investigate the paradox of the rs9939609 association with both obesity and lean mass.

I like the idea of using CRISPR-guided prime editing to perform allele substitution in an isogenic background in hESCs. I also endorse the leveraging of hESCs' potential to be differentiated to various cell types. Of course, care must be exercised as the in vitro differentiation protocols do not recapitulate all aspects of in vivo biology, but the authors appear to be aware of the caveats.

I must, however, raise a few major concerns that need to be resolved. 1.) the SNP the authors choose to focus on, rs9939609, is an index GWAS SNP, tightly linked to other, potentially causal SNPs that seem to have been ignored in the paper. In fact, one of the linked SNPs – rs1421085 – has previously been shown to be causal (Claussnitzer papers), and rs9939609 was excluded as a causal SNP (albeit that finding was reported in adipocytes). If the rs9939609 locus is gene-regulatory in muscle tissue but not in other tissues, then that is potentially very interesting, and could be expanded on, but more data is necessary (e.g. epigenetic marks). Also, if the rs9939609

SNP is a causal eQTL, I wonder how it compares to the other, tightly linked, SNP alleles in terms of effect size or tissue-specific gene regulation. The authors have not performed any such analysis, so at best one can only claim the rs9939609 to be one causal SNP; however, the net effect of all SNP alleles in the haplotype is what matters, and needs to be investigated somehow.

Another concern is the method of generating the cells with the different genotypes – were they generated using a similar protocol? That could have affected the results, although I do appreciate that the phenotype of the A allele carriers (n=6), as presented in the manuscript, is indeed striking. Also, for some reason only one control sample is shown in many figures versus six allele-edited samples, which is not a good praxis.

Please see my more detailed comments below.

Major points:

1. This is the biggest concern and must be addressed. The SNP of interest, rs9939609, is present on a 44 kb haploblock (according to a lookup in Haploreg) and is linked ($r^2 \geq 0.8$) to many other, potentially causal/functional SNPs. The authors state in the introduction that “every one of the dozens of FTO SNPs is in strong linkage disequilibrium with dozens of other SNPs”, but despite that choose to focus on only rs9939609. From my understanding, rs9939609 is a GWAS index SNP, used to represent the haploblock in question. This means any of the other SNPs present on that haploblock might actually be functional. As the minor alleles will all be inherited together, if any of them is causal and impacts a more pronounced phenotype than rs9939609-A allele, then the relevance of rs9939609 becomes lower, which affects the interpretation of the observed human obesity/lean mass association with the haplotype. Furthermore, the other SNPs might be active in different, non-muscle, tissues too. Looking at epigenetic marks (e.g. H3K27ac, ATAC-Seq) and transcription factor motif disruption could help to fine map any other causal SNP. Another idea would be to perform luciferase reporter assays for the linked SNPs in a relevant cell line to exclude their functionality.

2. If the rs9939609 is indeed causal, and more functional in muscle than in other tissues, one could look up and compare any epigenetic or chromatin features (in e.g. Encode, Roadmap epigenetics) to see if the rs9939609 locus has any muscle tissue-specific marks.

3. Were the non-edited cells (T/T genotype) obtained through the same treatment with prime editing reagent (or mock) transfection and single cell cloning? If not, that could have affected the results as

single cell cloning comes at the cost of a much higher passage number that can alter gene expression (e.g. of FTO), or some of the phenotypes presented in the manuscript (proliferation, myotube formation). To ensure comparability, the experiments need to be performed with cells that have passed through similar transfection and cloning procedure. In the methods section the authors state: “Six FTOs9939609-A hESC lines and their isogenic hESC control lines, all cultured in parallel, were used in all profiling and perturbation experiments”. I am not sure if “in parallel” includes single cell cloning and mock transfections, or just the differentiation+functional experiments. At the least, one could show that single-cell clones of hESCs (TT genotypes) have similar myogenic differentiation properties as the original (not cloned, with lower passage number) hESCs. Yet another way, where cloning could be avoided altogether (depending on editing efficiency of course), would be to perform a small deletion (20-30 bp or so) around rs9939609 using two Cas9+gRNAs and show that it recapitulates some of the critical phenotypes obtained through prime editing.

4. In most figures, especially in western blot images, there is a peculiar presentation of only one sample (!) from hESCs (T/T genotype) versus six samples of prime-edited cells (T/A or A/A). Why? Also, this further raises my concern that only one cell line of TT genotype was used, and compared with six TA/AA single cell clones. Even if you use technical replicates of one cell line, that cannot be reliably compared with single cell clones with much higher passage number and inter-clonal variation. The very small SEM on the hESCs in many figures, contrasting with the larger SEM of the FTO samples, appears to confirm my concerns. If I missed something, then please clarify.

Minor points:

1. The title is slightly misleading. It is not the knock-in that explains anything, but the effects of the knock-in. I think the title should be re-written to capture how the genotype explains the phenotype.
2. The introduction could be shortened to one page (or so); at least I find the introduction excessive as it stands.
3. Similarly, the discussion is very long; can it be written more concisely?
4. Throughout the paper the genotypes of the cells are referred to as FTOs9939609-TT or FTOs9939609-A . I don't understand why a single A is used, and not TA and AA, if the genotype is heterozygous and homozygous, respectively? Is there a reason to not consider allele dose-dependent effects of the A allele and look at TA and AA separately?
5. In GWAS Catalog, it seems that some FTO SNPs (albeit not rs9939609) are also associated with physical activity, which of course could affect BMI and lean body mass. Any thoughts on that? E.g. could FTO act on appetite or motivation for physical activity? Please discuss, and provide references/data if possible.

6. How does this research relate to the findings by Claussnitzer et al (NEJM 2015)? In the discussion it's mentioned that the A/A genotype does not lead to altered IRX3, but I think that is due to different mechanism regulated by the rs1421085 SNP identified as causal by Claussnitzer et al. This should be mentioned. Similar for the RPGRIP1L, if the mechanism is known.

7. Results, page 3: H1, H7, and H9 hESCs are mentioned – although those lines are commonly used, a reference and a brief description of them should be added to the methods.

8. For people interested in prime editing, please provide the allele editing efficiency achieved, i.e. how many clones out of all clones screened were A/A homozygous, without random indels? Also, if different prime editing vectors were tested, carrying different primer-binding sequences and reverse transcriptase templates, please add that to the methods section and/or to supplemental data.

9. Also, related to the point above, in the methods: “Genome editing of hESCs” the authors mention a Supplementary Table with listed pegRNAs and primers. I could not find that table in the files I have received (neither in merged pdf nor with supplemental figures), but maybe I missed something.

10. Data for Fig. 6: While I welcome the use of inhibitors to prove a specific mechanism, I am not sure how specific bisantrene and entacapone are for FTO? Any data on that? Otherwise, you could be looking at FTO inhibition as well as other, confounding effects.

11. Is there are reason authors use SEM and not standard deviation in their statistical analyses?

12. In the figures the different cells are referred to as hESCs and FTO. I would suggest to refer to the groups by their genotypes (TT vs. TA+AA). If there is any allele dose dependent additive effect I would actually prefer to show that, and redo the statistical analysis. Alternatively, if a dominant effect of the A allele was observed then please state that in the text to warrant the current plot design.

13. Font size of some graph labels is very small and could be increased to match the size of other labels, e.g. Fig 4e,f graphs (not the blots), and other graphs.

Textual concerns:

Page 1:

Quote: “to cleanly dissect and unmask potentially hidden roles of an edited FTO SNP during the black box process of human tissue and organ development.”

The phrases “cleanly dissect”, “black box” could be removed.

Point-by-Point Response to Reviewer Comments

Reviewer #1 (Remarks to the Author):

This is an interesting report addressing a pertinent research questions with novel observations that could transform our understanding of the role of FTO. There is an unmet need for insight into the mechanisms that regulate sarcopenia and this paper adds to our understanding. However, some concerns need to be addressed:

Skeletal muscle growth is reliant on a harmonious proliferation of diverse cell types, not solely the myogenic progenitors explored in this manuscript. Data from the Rando lab has underscored the pivotal role of mesenchymal progenitors in muscle growth (PMID: 31091443). How does this information align with the relatively modest effects observed in adipose progenitors? What impact does FTO^{9939609-A} have on PDGFR α positive cells? The isolated impact on myogenic progenitors might suggest a crucial function in recovering from muscle injuries, potentially providing defense against sarcopenia. What existing knowledge exists regarding recovery from muscle injuries in FTO transgenic models?

=> **We thank the Reviewer for these suggestions, our adipose progenitor cells (new Fig S4a) are actually also PDGFR α positive mesenchymal progenitor cells (see PMID 32004493). Our results show that FTO^{9939609-A} only has a somewhat modest impact on the specification of human mesenchymal progenitor cells, every marker considered. We are showing below, our qPCR results for PDGFR α expression in FTO-A/TT mesenchymal progenitor cells. Note the CT was ~26.**

=> As for existing knowledge regarding muscle regeneration in FTO transgenic models, Wang et al. 2023 (now our reference 36) did find a positive correlation between the expression level of FTO and the percentage of type I muscle fibers. During muscle regeneration after injury, soleus injection of a selective inhibitor of the FTO m6A demethylase led to a decrease in the percentage of type I muscle fibers and the levels of Myh7, Myh1 and Tnnt3. Collectively, these results revealed that FTO promotes the formation of type I myofibers in mice, which is consistent with our findings, though we found a more extensive promotion of both type I and II myogenesis in human cells (new Fig S3a). We have clarified these in our Discussion.

The examinations into insulin resistance reveal a correlation between FTO^{9939609-A} and insulin signaling. Nevertheless, the insulin doses utilized could potentially activate the IGF1

receptor as well, introducing complexities into interpreting PI3k/Akt signaling. This should be acknowledged in the discussion. Numerous data sets have been unable to establish links between insulin signaling activity and glucose uptake. Should assertions be made regarding heightened insulin sensitivity, it will be imperative to observe modifications in insulin-triggered substrate metabolism.

=> The Reviewer is correct, we agree. Both insulin and IGF-1 are able to bind to and activate each other's receptors, albeit with reduced affinity, and both IR and IGF1R also elicit common downstream kinase signaling, leading to increased substrate protein phosphorylation and glucose uptake (Cai W et al., Domain-dependent effects of insulin and IGF-1 receptors on signaling and gene expression. *Nat Commun.* 8:14892, 2017, now our reference 80). But given time, excessive activation can cause insulin/IGF resistance to develop, leading to lower glucose uptake. To confirm this, we performed glucose uptake assays in addition to the immunoblots for kinase signaling to downstream substrate proteins. Our results showed that the FTO rs9939609-A mutation does decrease insulin-stimulated glucose uptake (new Fig 4d), further confirming our findings on FTO rs9939609-A inducing insulin resistance. We have rewritten our Results and Discussion accordingly.

Reviewer #2 (Remarks to the Author):

Guang et al used CRISPR prime editing to knock-in the FTO rs9939609-A SNP into isogenic hESCs, which were then differentiated into tissue models, to study potentially hidden roles of the edited FTO SNP during human tissue development. They found that among the 6 lineages tested, FTO rs9939609-A had an effect on the human skeletal muscle lineage where they report effects of muscle progenitor proliferation, differentiation, senescence and muscle insulin resistance.

I have some methodological queries and concerns which should be addressed to improve the quality and relevance of this work:

- The specific protocol used for myogenic differentiation is not very clear and only cross-referenced via a previous paper for the same group. Overall it also appears to have a limited efficiency in terms of differentiation and an unclear purity on the output (no purification steps). It is therefore critical to validate the findings across multiple (e.g 3) different protocols to differentiate hPSCs into muscle to make sure the effect is not protocol specific. As an example, Figure 1b: these cells are supposed to be myotubes but they have remarkably low levels of MyHC (MF20) and high levels of Pax3/7. This profile is more of late myoblasts.

=> We agree with the Reviewer that a good protocol is needed to get reliable results on myogenesis, as bad protocols might simply hide any positive effects of FTO due to sheer impurity levels. Many protocols have been attempted for mesoderm/muscle differentiation (now our references 105-112), and the best protocol in terms of myoblast percentage was selected. Most protocols led to cell death or impure differentiation across all cell-lines (well-known in the field, reviewed in Chien, Xi, Pyle, 2022, PMID 34973262). One protocol was not that efficient in myogenesis but still showed that FTO promotes myogenesis in general (Xi et al., 2017, reference 108; new Supplementary Fig S4b-c), while the selected protocol gave the highest percentage ~60-70% myoblasts and ~30-40% myocytes (Loh et al., 2016, reference 105),

and showed that FTO promotes myogenesis. Most hESC-directed myogenesis protocols do not perform purification/sorting during differentiation, as the ECM cues needed to guide myogenesis would be lost. Fig 1b is actually a bar graph of FTOs9939609-A cells' relative gene expression profile, relative to FTOs9939609-TT hESC-derived cells. And Fig 1d is a semi-quantitative Western that is more for comparing between samples' protein levels, than for comparing between genes/proteins. The correct figures that depict MyHC (MF20) protein expression and differentiation in FTOs9939609-A myotubes, relative to FTOs9939609-TT hESC-derived myotubes, is Fig 2a-b. One can see most of the cells are MyHC-positive, there is typically 30-40% fusion in FTOs9939609-TT hESC-derived myotubes, and >50% fusion in FTOs9939609-A myotubes.

- Please show the individual data of each clone (i.e. for H1, H7 and H9 hESCs) across all experiments to better quantify the variability across the different lines

=> **We thank the Reviewer for this suggestion. We did not display data on the FTOs9939609-TT H7 and H9 hESCs previously, because they were so similar to FTOs9939609-TT H1 in phenotype, and we wanted the reader to focus on the variability of FTOs9939609-A gene editing, by comparing to one FTOs9939609-TT control to stay focused. We have now separately included data comparing between FTOs9939609-TT H1, H7 and H9 hESC clones in Supplementary Fig S3j-m, Fig S4e-h, Fig S5a-b, Fig S7b-e,h-p.**

- I assume the controls here have also been subjected to the same clonal procedure that the edited cells have undergone?

=> **Yes, single cell cloning was done in parallel with the same number of passages for all our control FTOs9939609-TT hESC lines (H1, H7, H9, all of which were very similar) and FTOs9939609-A lines, to ensure comparability between the FTOs9939609-TT lines and the FTOs9939609-A lines.**

Reviewer #3 (Remarks to the Author):

In this paper the authors use human embryonic stem cells in which they perform prime editing of the SNP rs9939609 that associates with obesity but also with lean mass. The stem cells are then differentiated to six different cell types, and a phenotypic effect of the allele editing is observed in skeletal muscle-like cells, in which functional assays are performed to investigate the paradox of the rs9939609 association with both obesity and lean mass.

I like the idea of using CRISPR-guided prime editing to perform allele substitution in an isogenic background in hESCs. I also endorse the leveraging of hESCs' potential to be differentiated to various cell types. Of course, care must be exercised as the in vitro differentiation protocols do not recapitulate all aspects of in vivo biology, but the authors appear to be aware of the caveats.

I must, however, raise a few major concerns that need to be resolved. 1.) the SNP the authors choose to focus on, rs9939609, is an index GWAS SNP, tightly linked to other, potentially causal

SNPs that seem to have been ignored in the paper. In fact, one of the linked SNPs – rs1421085 – has previously been shown to be causal (Claussnitzer papers), and rs9939609 was excluded as a causal SNP (albeit that finding was reported in adipocytes). If the rs9939609 locus is gene-regulatory in muscle tissue but not in other tissues, then that is potentially very interesting, and could be expanded on, but more data is necessary (e.g. epigenetic marks). Also, if the rs9939609 SNP is a causal eQTL, I wonder how it compares to the other, tightly linked, SNP alleles in terms of effect size or tissue-specific gene regulation. The authors have not performed any such analysis, so at best one can only claim the rs9939609 to be one causal SNP; however, the net effect of all SNP alleles in the haplotype is what matters, and needs to be investigated somehow.

Another concern is the method of generating the cells with the different genotypes – were they generated using a similar protocol? That could have affected the results, although I do appreciate that the phenotype of the A allele carriers (n=6), as presented in the manuscript, is indeed striking. Also, for some reason only one control sample is shown in many figures versus six allele-edited samples, which is not a good praxis.

=> We thank the Reviewer for this suggestion. We did not display data on the FTOs9939609-TT H7 and H9 hESCs previously, because they were so similar to FTOs9939609-TT H1 in phenotype, and we wanted the reader to focus on the variability of FTOs9939609-A gene editing, by comparing to one FTOs9939609-TT control to stay focused. We have now separately included data comparing between FTOs9939609-TT H1, H7 and H9 hESC clones in Supplementary Fig S3j-m, Fig S4e-h, Fig S5a-b, Fig S7b-e,h-p.

Please see my more detailed comments below.

Major points:

1. This is the biggest concern and must be addressed. The SNP of interest, rs9939609, is present on a 44 kb haploblock (according to a lookup in Haploreg) and is linked ($r^2 \geq 0.8$) to many other, potentially causal/functional SNPs. The authors state in the introduction that “every one of the dozens of FTO SNPs is in strong linkage disequilibrium with dozens of other SNPs”, but despite that choose to focus on only rs9939609. From my understanding, rs9939609 is a GWAS index SNP, used to represent the haploblock in question. This means any of the other SNPs present on that haploblock might actually be functional. As the minor alleles will all be inherited together, if any of them is causal and impacts a more pronounced phenotype than rs9939609-A allele, then the relevance of rs9939609 becomes lower, which affects the interpretation of the observed human obesity/lean mass association with the haplotype. Furthermore, the other SNPs might be active in different, non-muscle, tissues too. Looking at epigenetic marks (e.g. H3K27ac, ATAC-Seq) and transcription factor motif disruption could help to fine map any other causal SNP. Another idea would be to perform luciferase reporter assays for the linked SNPs in a relevant cell line to exclude their functionality.

=> Our working hypothesis was that at least one of the FTO SNPs might be a causal eQTL that influences nearby enhancer activity. We betted on rs9939609 because it is one of the most well-studied FTO SNPs (over 700 papers in Pubmed), had one of the strongest P-values (as mentioned in the Introduction), and scanning of genomics databases revealed that it was a potential eQTL, especially in the skeletal muscles (new Supplementary Fig S7f-g):

As recommended, close examination of the epigenetic H3K27ac peaks near rs9939609 (Encode), revealed an FTO enhancer EH38E181645 that lies within 1kb of rs9939609, and it is a cCRE (cis-Regulatory Element) with peaks in DNase hypersensitivity, H3K4me3 enrichment, H3K27ac enrichment, and CTCF binding, especially in muscle cells (our new Fig 5e):

In fact, we were pleasantly surprised to see that a recent paper (Andersen et al., 2023; PMID 36809463, our reference 42) showed a physical interaction between the FTO promoter and an enhancer region encompassing rs9939609, and that the rs9939609-A allele correlated with higher *FTO* expression ($P=0.011$), although no further molecular and cellular studies were performed.

To experimentally determine if this FTO enhancer is affected by allelic differences in FTO rs9939609, we performed H3K27ac CHIP-qPCR on all rs9939609-TT vs rs9939609-A lines, for all 5 tissue types. Our results showed that our CRISPR-mediated FTO rs9939609-A mutation caused a significant increase in H3K27ac at this FTO enhancer EH38E181645 in skeletal muscle cells (new Fig 5f). In contrast, adipocytes only showed a mild increase in H3K27ac at this FTO

enhancer EH38E1816455, and little to no differences in other cell types (new Fig S7h-k). All three hESC lines' derived cells showed insignificant line-to-line variation in H3K27ac at this FTO enhancer EH38E1816455 (new Fig S7h-p). We have updated our Results to reflect these new findings.

2. If the rs9939609 is indeed causal, and more functional in muscle than in other tissues, one could look up and compare any epigenetic or chromatin features (in e.g. Encode, Roadmap epigenetics) to see if the rs9939609 locus has any muscle tissue-specific marks.

=> **As recommended by the Reviewer, we examined a variety of databases and references to check if our findings made sense at the epigenomic level.**

According to the GTEx, FUSION, Encode, and Roadmap databases, which held data for SNPs, ChIP-seq, whole genome bisulfite sequencing (WGBS), and gene expression (RNAseq), and according to a paper that examined GWAS for lean mass (Nat Comm 2017, our reference 37), both rs9939609 and rs11649091 were correlated with *FTO* expression in skeletal muscles, but not in the adipose or liver tissue. Moreover, in the GTEx database, there were clearly no eQTLs near FTO rs9939609 in the heart, brain nor other tissues, in contrast to skeletal muscle tissue.

And as shown above, the Encode database shows an enhancer within +1kb from rs9939609: EH38E1816455, a cCRE with DNase, H3K4me3, H3K27ac, and CTCF peaks in skeletal muscle cells (new Fig 5e).

3. Were the non-edited cells (T/T genotype) obtained through the same treatment with prime editing reagent (or mock) transfection and single cell cloning? If not, that could have affected the results as single cell cloning comes at the cost of a much higher passage number that can alter gene expression (e.g. of *FTO*), or some of the phenotypes presented in the manuscript (proliferation, myotube formation). To ensure comparability, the experiments need to be performed with cells that have passed through similar transfection and cloning procedure. In the methods section the authors state: "Six FTO rs9939609-A hESC lines and their isogenic hESC control lines, all cultured in parallel, were used in all profiling and perturbation experiments". I am not sure if "in parallel" includes single cell cloning and mock transfections, or just the differentiation+functional experiments. At the least, one could show that single-cell clones of hESCs (TT genotypes) have similar myogenic differentiation properties as the original (not cloned, with lower passage number) hESCs. Yet another way, where cloning could be avoided altogether (depending on editing efficiency of course), would be to perform a small deletion (20-30 bp or so) around rs9939609 using two Cas9+gRNAs and show that it recapitulates some of the critical phenotypes obtained through prime editing.

=> **We agree with the Reviewer's concerns. In fact, single cell cloning was done in parallel with the same low number of passages for all three of our control FTO rs9939609-TT hESC lines (H1, H7, H9, all of which were very similar) and FTO rs9939609-A lines, to ensure comparability between the rs9939609-TT lines and rs9939609-A lines.**

As for a small deletion, we do not think that would recapitulate the phenotypes we see for two reasons. Firstly, our edited FTO rs9939609-A allele seems to be promoting FTO enhancer

H3K27ac, FTO expression and FTO activity, not disrupting FTO activity, whereas a deletion is usually more likely to disrupt function. Secondly, our study is focused on the rs9939609 SNP (building on the corpus of literature on this SNP), not an entire stretch of DNA around the SNP. It is possible that even a small deletion could have unpredictable and artifactual consequences for all the enhancers around the *FTO* locus.

4. In most figures, especially in western blot images, there is a peculiar presentation of only one sample (!) from hESCs (T/T genotype) versus six samples of prime-edited cells (T/A or A/A). Why? Also, this further raises my concern that only one cell line of TT genotype was used, and compared with six TA/AA single cell clones. Even if you use technical replicates of one cell line, that cannot be reliably compared with single cell clones with much higher passage number and inter-clonal variation. The very small SEM on the hESCs in many figures, contrasting with the larger SEM of the FTO samples, appears to confirm my concerns. If I missed something, then please clarify.

=> **We thank the Reviewer for these suggestions. To clarify further, one T/A line and one A/A line were generated from each isogenic FTOs9939609-TT line (H1 or H7 or H9) using the same protocol. They all went through the exact same (reagent or mock) transfection, single-cell cloning, expansion over same low number of passages, differentiation over same number of days etc, in parallel. We did not display all the data on the FTOs9939609-TT H7 and H9 hESCs previously, because they were so similar in phenotype (though they are 3 hESC lines independently derived from different blastocysts, as clarified by the Methods and now references 102-104), and we wanted the reader to focus on the variability of FTO rs9939609-A gene editing. The larger SEM of the FTO rs9939609-A samples is precisely due to the inherent variance of CRISPR-edited lines, whereas the unedited FTO rs9939609-TT H1 or H7 or H9 lines are much more similar to each other. We have now separately included data comparing rs9939609-TT H1, H7 and H9 in Supplementary Fig S3j-m, Fig S4e-h, Fig S5a-b, Fig S7b-e,h-p.**

Minor points:

1. The title is slightly misleading. It is not the knock-in that explains anything, but the effects of the knock-in. I think the title should be re-written to capture how the genotype explains the phenotype. => **We thank the Reviewer for this suggestion. We have re-worded the title to, “An obesogenic FTO allele causes accelerated development, growth and insulin resistance in human skeletal muscle cells”.**

2. The introduction could be shortened to one page (or so); at least I find the introduction excessive as it stands. => **We thank the Reviewer for this suggestion, we have shortened the Introduction to about one page, as recommended.**

3. Similarly, the discussion is very long; can it be written more concisely? => **We thank the Reviewer for this suggestion, we have shortened the Discussion as recommended.**

4. Throughout the paper the genotypes of the cells are referred to as FTOs9939609-TT or FTOs9939609-A . I don't understand why a single A is used, and not TA and AA, if the genotype is heterozygous and homozygous, respectively? Is there a reason to not consider allele dose-dependent effects of the A allele and look at TA and AA separately? - => **We thank the Reviewer for this suggestion. Indeed, dose-dependent effects were not observed with TA vs AA in all**

cellular phenotypes, so we decided to present data as rs9939609-A as if its dominant (except raw immunoblots). We have clarified the Results section accordingly.

5. In GWAS Catalog, it seems that some FTO SNPs (albeit not rs9939609) are also associated with physical activity, which of course could affect BMI and lean body mass. Any thoughts on that? E.g. could FTO act on appetite or motivation for physical activity? Please discuss, and provide references/data if possible.

=> Indeed, obesity-associated FTO SNPs were found to be associated with increased energy intake (Cecil et al., 2008; Speakman et al., 2008; Timpson et al., 2008), increased intake of dietary fat (Park et al., 2013; Timpson et al., 2008), increased appetite or reduced satiety (Wardle et al., 2008; Goltz et al., 2019), and loss of control over eating (Tanofsky et al., 2009). These correlations are consistent with an increased BMI, but their cause-effect relationships with increased lean mass had remained unclear. In fact, the positive association of human FTO rs9939609-A with both fat mass and lean mass is paradoxical, because studies have indicated that lean mass is negatively correlated with obesity, IR and MetS.

Many studies have also consistently shown that FTO SNPs are not associated with physical activity levels (Ahmad et al., 2010; Franks et al., 2008; Speakman et al., 2008; Vimalaswaran et al., 2009; Loos et al., 2014; Kirac et al., 2016). However, FTO's effect on obesity susceptibility is attenuated by approximately 30-47% in physically active individuals (Loos et al., 2014; Andersen et al., 2023), and physical activity is an independent variable (in parallel with FTO SNP status) that influences BMI. We have incorporated these as references 42, 47-58 and included them in our Discussion.

6. How does this research relate to the findings by Claussnitzer et al (NEJM 2015)? In the discussion it's mentioned that the A/A genotype does not lead to altered IRX3, but I think that is due to different mechanism regulated by the rs1421085 SNP identified as causal by Claussnitzer et al. This should be mentioned. Similar for the RPGRIP1L, if the mechanism is known.
=> We agree with the Reviewer's assessment, the differences are most likely due to the different mechanisms of different SNPs, which are active in different cell-types. In Claussnitzer et al. 2015 (reference 23), the rs1421085 T-to-C single-nucleotide variant disrupts a conserved motif for the ARID5B repressor in adipose progenitors, which led to derepression of a potent preadipocyte enhancer, thereby doubling *IRX3* and *IRX5* expression during early adipocyte differentiation. In Stratigopoulos et al. 2014, the rs8050136-A allele disrupted a neuronal CUX1 regulatory element, leading to reduced *RPGRIP1L* expression, which increased feeding-related peptides by 30-70% in the hypothalamus. In contrast, in our studies, the rs9939609 T-to-A variant strengthened H3K27 acetylation at the nearby muscle-specific FTO enhancer EH38E1816455 to triple *FTO* expression, which demethylated and increased *H19/IGF2* RNAs and pro-myogenic factors by several- to thousands-fold during muscle fiber maturation. We have rewritten our Results and Discussion to clarify these interpretations of earlier publications and our data.

Claussnitzer M, et al. FTO Obesity Variant Circuitry and Adipocyte Browning in Humans. *N Engl J Med.* 2015 Sep 3;373(10):895-907.

Stratigopoulos G, et al. Hypomorphism for RPGRIP1L, a ciliary gene vicinal to the FTO locus, causes increased adiposity in mice. *Cell Metab.* 2014 May 6;19(5):767-79.

7. Results, page 3: H1, H7, and H9 hESCs are mentioned – although those lines are commonly used, a reference and a brief description of them should be added to the methods. => **Ok, references and brief descriptions of the hESC lines have been added to the Methods section.**
8. For people interested in prime editing, please provide the allele editing efficiency achieved, i.e. how many clones out of all clones screened were A/A homozygous, without random indels? Also, if different prime editing vectors were tested, carrying different primer-binding sequences and reverse transcriptase templates, please add that to the methods section and/or to supplemental data. => **We have added the information to the Methods section as suggested.**
9. Also, related to the point above, in the methods: “Genome editing of hESCs” the authors mention a Supplementary Table with listed pegRNAs and primers. I could not find that table in the files I have received (neither in merged pdf nor with supplemental figures), but maybe I missed something. => **Apologies, we made sure the Supplementary Table shows up this time.**
10. Data for Fig. 6: While I welcome the use of inhibitors to prove a specific mechanism, I am not sure how specific bisantrene and entacapone are for FTO? Any data on that? Otherwise, you could be looking at FTO inhibition as well as other, confounding effects.

=> **Su et al. 2020 (reference 45) conducted a screen and assessed the top compounds' efficacy on inhibition of FTO's m6A demethylase activity through cell-free m6A demethylase assays. CS1 (NSC337766, also named Bisantrene) displayed robust inhibition of FTO's demethylase activity (their Fig. 1E). They also identified CS1/Bisantrene's robust inhibition of FTO in MONOMAC 6 cells (their Fig. 2J). Their docking models suggest that CS1/Bisantrene binds tightly to FTO protein and blocks its catalytic pocket (their Fig. 1G). Additionally, based on the crystal structure of an FTO-oligonucleotide complex, they found that CS1/Bisantrene interacts with FTO residues that were known to be involved in the binding of FTO with m6A modified ssDNA, such as HIS231 and GLU234.**

These data suggest that CS1/Bisantrene selectively binds to and occupies the catalytic pocket of FTO, and thereby blocks m6A-modified oligos from entering FTO's catalytic pocket, which in turn inhibits FTO's demethylase activity on target RNA transcripts.

Su, R. et al. Targeting FTO suppresses cancer stem cell maintenance and immune evasion. *Cancer Cell* 38, 79-96. e11 (2020).

Peng et al. 2019 (reference 46) found that Entacapone inhibited FTO demethylation activity at a median inhibitory concentration (IC50) of 3.5 μM (their Fig. 1B). Consistent with the effect of FTO knockdown, entacapone treatment also enhanced the amount of m6A on mRNA in a

variety of human cell lines, including Hep-G2 (their Fig. 1C). In contrast, entacapone did not show any inhibitory effect on the enzymatic activity of the RNA m⁶A demethylase AlkB homolog 5 (ALKBH5) nor the ten-eleven translocation methylcytosine dioxygenase 1 (TET1), nor did it alter the DNA methylation or histone methylation patterns in entacapone-treated HepG2 cells (their Fig. S1D-G).

Peng et al. Identification of entacapone as a chemical inhibitor of FTO mediating metabolic regulation through FOXO1. *Science Translational Medicine* 11, eaau7116 (2019).

Both papers detailing the specific inhibition of FTO m⁶A methylase, are now cited in our Results (references 45-46).

11. Is there are reason authors use SEM and not standard deviation in their statistical analyses? => We are following the standard practice in cellular and molecular biochemistry for depicting SEM instead of SD to portray the variability, no other special statistical reasons.

12. In the figures the different cells are referred to as hESCs and FTO. I would suggest to refer to the groups by their genotypes (TT vs. TA+AA). If there is any allele dose dependent additive effect I would actually prefer to show that, and redo the statistical analysis. Alternatively, if a dominant effect of the A allele was observed then please state that in the text to warrant the current plot design. => Unfortunately, no dose-dependent effects were observed in our cellular and molecular phenotypes. Instead, a dominant effect of the A allele was observed, which is why we labeled the plots as such. We have clarified this more clearly in the Results as suggested.

13. Font size of some graph labels is very small and could be increased to match the size of other labels, e.g. Fig 4e,f graphs (not the blots), and other graphs. => **No problem, amended as suggested.**

Textual concerns:

Page 1:

Quote: "to cleanly dissect and unmask potentially hidden roles of an edited FTO SNP during the black box process of human tissue and organ development."

The phrases "cleanly dissect", "black box" could be removed.

=> **No problem, amended as suggested.**

REVIEWER COMMENTS

Reviewer #1 (Remarks to the Author):

The revisions have improved the quality of the manuscript.

I only have minor concerns. The prominent role of skeletal muscle in development of insulin resistance (ll 426-427) should be more clear, and papers like Warram et al should be acknowledged: doi: 10.7326/0003-4819-113-12-909

Reviewer #3 (Remarks to the Author):

I find the following statement in their rebuttal rather odd, although I think I know what they mean: “We did not display data on the FTOs9939609-TT H7 and H9 hESCs previously, because they were so similar to FTOs9939609-TT H1 in phenotype, and we wanted the reader to focus on the variability of FTOs9939609-A gene editing, by comparing to one FTOs9939609-TT control to stay focused.” (emphasis mine)

But a potentially worrying issue is what’s stated in lines 143-144: “it was clearer to compare the variability of FTOs9939609-A gene editing to one representative FTOs9939609-TT isogenic hESC clone that was cultured in parallel (FTO-TT).” In fact, it seems the authors used only a single control TT clone in their experiments against several edited A- clones, which is a very odd practice. I welcome that the authors have now added the control cell line data to supplemental figures, so the readers themselves can assess the variation in those cell lines, but they seem to have been tested in separate assays, without the A- clones. Is that correct?

I also asked: “For people interested in prime editing, please provide the allele editing efficiency achieved, i.e. how many clones out of all clones screened were A/A homozygous, without random indels? Also, if different prime editing vectors were tested, carrying different primer-binding sequences and reverse transcriptase templates, please add that to the methods section and/or to supplemental data.” And received the reply “We have added the information to the Methods section as suggested.” Well, I still don’t see the allele editing efficiency reported in the Methods section, and if it's not available then please state that clearly.

I welcome the new data regarding the causality of rs9939609, but the exclusion of other SNPs as causal is not experimentally substantiated. I would say the best interpretation of the data would still be that rs9939609 is a causal SNP, but contribution of other SNPs in the haplotype block cannot be excluded. This should definitely be mentioned in the Discussion. The authors' argument, in the rebuttal letter, that: "We betted on rs9939609 because it is one of the most well-studied FTO SNPs (over 700 papers in Pubmed), had one of the strongest P-values (as mentioned in the Introduction), and scanning of genomics databases revealed that it was a potential eQTL, especially in the skeletal muscles " does not exclude the potential causality/functionality of other SNPs, because all the mentioned data are actually correlating an entire haplotype, not just the single SNP. On the other hand, I do appreciate that the allele editing (T to A) appears to alter the cell phenotype markedly, and the A allele appears to be causing that change, which is nice.

POINT BY POINT RESPONSE TO ALL REVIEWER COMMENTS

Reviewer #1 (Remarks to the Author):

The revisions have improved the quality of the manuscript.

I only have minor concerns. The prominent role of skeletal muscle in development of insulin resistance (ll 426-427) should be more clear, and papers like Warram et al should be acknowledged: doi: 10.7326/0003-4819-113-12-909

=> We thank you for this suggestion. We have cited the Warram et al. paper in our Discussion, “Warram et al revealed that one to two decades before type 2 diabetes is diagnosed, reduced glucose clearance is already present. This reduced clearance is accompanied by compensatory hyperinsulinemia, not hypoinsulinemia, suggesting that the primary defect is in peripheral tissue response to insulin and glucose, not in the pancreatic beta cell¹⁰²”, to clarify the early indications of a prominent role of skeletal muscle in development of insulin resistance during T2D pathogenesis.

Reviewer #3 (Remarks to the Author):

I find the following statement in their rebuttal rather odd, although I think I know what they mean: “We did not display data on the FTOs9939609-TT H7 and H9 hESCs previously, because they were so similar to FTOs9939609-TT H1 in phenotype, and we wanted the reader to focus on the variability of FTOs9939609-A gene editing, by comparing to one FTOs9939609-TT control **to stay focused.**” (emphasis mine)

But a potentially worrying issue is what’s stated in lines 143-144: “it was clearer to compare the variability of FTOs9939609-A gene editing to one representative FTOs9939609-TT isogenic hESC clone that was cultured in parallel (FTO-TT).” In fact, it seems the authors used only a single control TT clone in their experiments against several edited A- clones, which is a very odd practice. I welcome that the authors have now added the control cell line data to supplemental figures, so the readers themselves can assess the variation in those cell lines, but they seem to have been tested in separate assays, without the A- clones. Is that correct?

=> We thank you for these suggestions. In fact, our experiments that used the control hESC - TT clones from all three hESC lines and the edited hESC-A clones from all three hESC lines were performed in parallel. We regret showing the plots/blots separately for the sake of clarity/emphasis. As recommended by the Editor we have now plotted all the data points for the control hESC -TT clones and the edited hESC-A clones together on the same graphs in different colours. H1 hESC line coloured in purple, H7 hESC line coloured in light blue and H9 hESC line coloured in dark blue.

I also asked: “For people interested in prime editing, please provide the allele editing efficiency achieved, i.e. how many clones out of all clones screened were A/A homozygous, without random indels? Also, if different prime editing vectors were tested, carrying different primer-binding sequences and reverse transcriptase templates, please add that to the methods section and/or to supplemental data.” And received the reply “We have added the information to the

Methods section as suggested.” Well, I still don’t see the allele editing efficiency reported in the Methods section, and if it's not available then please state that clearly.

=> Apologies for the accidental omission during the revision process, we have made doubly sure the information is added in the Methods section in red this time. In total, we confirmed 3 homozygous clones, after screening a large number with the first version of CRISPR prime editing.

I welcome the new data regarding the causality of rs9939609, but the exclusion of other SNPs as causal is not experimentally substantiated. I would say the best interpretation of the data would still be that rs9939609 is a causal SNP, but contribution of other SNPs in the haplotype block cannot be excluded. This should definitely be mentioned in the Discussion. The authors’ argument, in the rebuttal letter, that: “We betted on rs9939609 because it is one of the most well-studied FTO SNPs (over 700 papers in Pubmed), had one of the strongest P-values (as mentioned in the Introduction), and scanning of genomics databases revealed that it was a potential eQTL, especially in the skeletal muscles ” does not exclude the potential causality/functionality of other SNPs, because all the mentioned data are actually correlating an entire haplotype, not just the single SNP. On the other hand, I do appreciate that the allele editing (T to A) appears to alter the cell phenotype markedly, and the A allele appears to be causing that change, which is nice.

=> We thank you for these suggestions. We have added this best interpretation of the data to the end of the first paragraph of the Discussion section in red, “While the best interpretation of our data would be that rs9939609 is a causal SNP, the contribution of other SNPs in the haplotype block cannot be excluded”.